# CVSearch: Empowering Multimodal LLMs with Cognitive Visual Search for High-Resolution Image Perception

Liupeng Li [1 2]   Haoqian Kang [1]   Zhenyu Lu [2 3]   Jinpeng Wang [1]
Bin Chen [1]   Ke Chen [2]   Yaowei Wang [1 2]

## Abstract

High-resolution (HR) image perception presents a key bottleneck for multimodal large language models (MLLMs). While visual search offers a promising solution, existing methods struggle with the trade-off between coverage and efficiency. Visual expert-assisted search is efficient but prone to blind spots when proposals fail, whereas scan-based search guarantees coverage at the cost of computational redundancy and semantic fragmentation. To address this dilemma, we introduce *CVSearch*, a training-free adaptive framework that dynamically schedules search strategies via an *Assess-then-Search* workflow. Specifically, *CVSearch* first invokes expert-assisted search when global information is insufficient, and only triggers a novel semantic-aware scanning mechanism upon failure. Distinct from rigid grid partitioning, this efficient scanning paradigm incorporates *Semantic Guided Adaptive Patching* to decompose images into semantically consistent regions, effectively mitigating object fragmentation. Furthermore, we devise a *Dynamic Bottom-Up Search* strategy driven by a *Visual Complexity* prior to enable efficient and precise iterative exploration of local details. Extensive experiments on HR benchmarks demonstrate that *CVSearch* achieves state-of-the-art accuracy while substantially improving search efficiency. Code is released at ICML26-CVSearch.

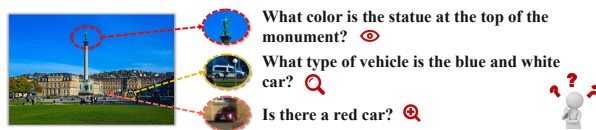

*(a)* Varying perceptual scales in real-world scenarios.

| | Method | | | | Efficiency |
|---|---|---|---|---|---|
| Direct Answer | -- | ✓ | ✗ | ✗ | High |
| Visual Expert Assisted | DyFo | ✓ | ✓ | ✗ | Medium |
| Scan-based Search | RAP | ✓ | ✓ | ✓ | Low |

*(b)* Comparison of different visual search modes.

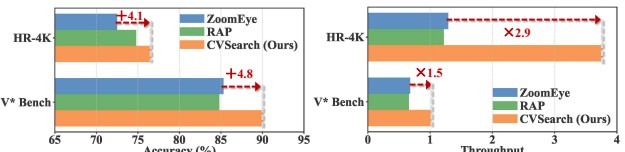

*(c)* Performance comparison on HR benchmarks.

*Figure 1.* (a) Real-world HR image perception requires handling targets with distinct granularities. (b) Existing methods struggle to balance coverage and efficiency. Visual expert assisted methods lack sufficient coverage for tiny targets, while scan-based methods ensure coverage but suffer from low efficiency. (c) Built upon Qwen2.5-VL-7B, *CVSearch* achieves the best balance, delivering SOTA accuracy with competitive throughput.

## 1. Introduction

The integration of Large Language Models (LLMs) (Touvron et al., 2023; Team et al., 2024) with visual encoders (Radford et al., 2021; Zhai et al., 2023) has revolutionized multimodal understanding, giving birth to Multimodal LLMs (MLLMs) capable of sophisticated reasoning. Despite this progress, current MLLMs largely rely on fixed-resolution processing schemes (Liu et al., 2024; Wang et al., 2024), which inevitably introduce a perceptual bottleneck. High-resolution (HR) images are aggressively downsampled, rendering the model blind to small objects and fine-grained details essential for real-world tasks (Zhang et al., 2024a), as illustrated in Figure 1(a).

To mitigate this limitation, recent research has branched into three paradigms: (1) Cropping-based paradigms (Li

---

[1]Harbin Institute of Technology, Shenzhen, China [2]Peng Cheng Laboratory, Shenzhen, China [3]Shenzhen Institutes of Advanced Technology, Chinese Academy of Sciences, Shenzhen, China. Correspondence to: Jinpeng Wang <wangjp26@gmail.com>, Yaowei Wang <wangyaowei@hit.edu.cn>.

*Proceedings of the 43rd International Conference on Machine Learning*, Seoul, South Korea. PMLR 306, 2026. Copyright 2026 by the author(s).

et al., 2024a;b;c) partition images into local crops. While this preserves details, it severs spatial coherence, causing semantic fragmentation where objects spanning crop boundaries are split into disjoint tokens. (2) HR Visual Encoder paradigms (Ge et al., 2024; Luo et al., 2025) inject high-frequency features via complex architectural modifications (hierarchical backbones or adapters) but struggle with varying aspect ratios. (3) Visual Search paradigms (Wu & Xie, 2024; Wang et al., 2025c; Shen et al., 2025; Lu et al., 2026a;b) represent a shift from passive processing to active perception, dynamically exploring relevant regions.

While visual search offers a promising alternative, existing approaches face a stark dichotomy between efficiency and robustness (see Figure 1(b)). **Visual Expert Assisted Search** (*e.g.,* SEAL (Wu & Xie, 2024), DyFo (Li et al., 2025a), $V^2$-SAM (Pan et al., 2025)) leverages external vision experts for rapid localization. While efficient, these systems are fragile because their performance is upper-bounded by the expert's capability. In scenarios involving tiny or occluded objects where the expert fails to generate accurate proposals, the MLLM is left with no fallback, leading to irreversible blind spots. Conversely, **Scan-based Visual Search** (*e.g.,* RAP (Wang et al., 2025d), ZoomEye (Shen et al., 2025), $DC^2$ (Wang et al., 2025c)) ensures exhaustive coverage through rigid grid scanning but is semantic-agnostic. These methods suffer from two critical drawbacks. They waste computation on information-sparse backgrounds due to uniform resource allocation, incurring prohibitive latency, and their rigid grid-based partitioning fractures object semantics, undermining downstream reasoning. This dichotomy presents a critical challenge: *How can we bridge the gap between the efficiency of expert guidance and the robustness of exhaustive search, without compromising semantic integrity?*

In this work, we propose **CVSearch**, a training-free framework that empowers MLLMs with cognitive, human-like visual search capabilities. Drawing inspiration from the cognitive process of human visual search (Wolfe et al., 2011; Li et al., 2025b), **CVSearch** implements a cognitive *Assess-then-Search* workflow, dynamically alternating between non-selective global perception (for gist extraction) and selective serial attention (for detailed scrutiny) based on task difficulty.

To resolve the efficiency-robustness dilemma, *CVSearch* introduces a hierarchical framework underpinned by three key innovations. Firstly, we propose a *Cognitive-Driven Adaptive Switching Mechanism* to dynamically schedules search modes of varying complexity. This mechanism mimics human cognitive control by prioritizing efficient visual expert assisted search (powered by SAM 3 (Carion et al., 2025)) upon detecting insufficient global information. Crucially, instead of treating expert failure as a dead-end, the

mechanism interprets it as a signal to activate the proposed *Scene-aware Scanning* mode, ensuring a seamless transition from rapid localization to comprehensive exploration. Secondly, within the *Scene-aware Scanning* phase, we introduce two complementary strategies to mitigate the limitations of conventional scanning. To address semantic fragmentation, we propose *Semantic Guided Adaptive Patching (SGAP)*. Capitalizing on the insight (Zou et al., 2023; Fu et al., 2025) that deep visual features from the expert retain rich scene semantics even when explicit localization fails, *SGAP* clusters these features to partition the image into semantically coherent regions rather than rigid grids. Simultaneously, it quantifies a *Visual Complexity Prior* to identify and prune redundant background branches, focusing computation on high-entropy areas. Furthermore, to overcome the error propagation inherent in top-down methods, we devise a *Dynamic Bottom-Up Search* strategy. By initiating exploration from information-dense leaf nodes and aggregating evidence upwards, this strategy not only ensures robust evidence collection but also enables an iterative search mechanism, allowing the model to refine its focus and recover from initial search failures.

Our contributions are summarized as follows:

- **Cognitive Hierarchical Framework:** We present **CVSearch**, the first training-free framework to unify the efficiency of visual expert assisted search with the robustness of semantic-aware scanning via a cognitive, failure-aware switching mechanism.
- **Semantic-Preserving Granularity:** We propose *(SGAP)*, which repurposes visual expert features to construct semantically consistent image patches, overcoming the fragmentation of rigid grids.
- **Robust Bottom-Up Exploration:** We introduce a dynamic bottom-up search strategy that prevents error propagation inherent in top-down methods, significantly improving small object perception.
- **SOTA Performance:** Extensive experiments on HR benchmarks demonstrate that **CVSearch** achieves state-of-the-art accuracy while substantially improving search efficiency compared to scan-based baselines.

## 2. Related Works

### 2.1. High-Resolution Image Perception in MLLMs

To bridge the gap between the limited input resolution of pre-trained vision encoders (*e.g.,* $336 \times 336$) and real-world demands for fine-grained details, existing strategies primarily fall into three paradigms. *Cropping-based methods* (Li et al., 2024a;b;c) partition high-resolution (HR) images into fixed grids. While preserving local details, they suffer from the "semantic sawtooth" effect (Huang et al., 2024), where rigid partitioning fractures objects across patches, disrupt-

ing semantic coherence. Furthermore, being semantically agnostic, they incur computational redundancy by processing empty backgrounds equally with dense foregrounds. *HR Visual Encoder* (Ge et al., 2024; Luo et al., 2025) mitigate token explosion via hierarchical backbones (*e.g.,* ConvNeXt (Woo et al., 2023)) or adaptors. However, they rely on global processing and lack the flexibility to selectively ignore irrelevant regions, often necessitating aggressive downsampling. *Visual Search frameworks* shift towards active perception. Some approaches (*e.g.,* SEAL (Wu & Xie, 2024), *DyFo* (Li et al., 2025a)) leverage external experts for region proposal, while others (*e.g.,* ZoomEye (Shen et al., 2025), *RAP* (Wang et al., 2025d)) employ tree-structured scanning. Despite progress, a critical trade-off remains: scan-based methods are robust but inefficient, while expert assisted methods are efficient but fragile upon expert failure. **CVSearch** resolves this dichotomy via a cognitive mechanism that intelligently switches between fast expert search and robust semantic-aware scanning, further enabled by bottom-up error correction.

## 2.2. Cognitive Mechanisms of Visual Search

Cognitive theories of human vision posit that visual search is not a unitary process but an interplay between two distinct pathways: a *selective pathway* and a *nonselective pathway* (Wolfe et al., 2011). The *nonselective pathway* rapidly extracts global gist in parallel to reject vast irrelevant regions. In contrast, the *selective pathway* performs serial, capacity-limited processing of specific objects guided by attentional templates (Wolfe, 2020). Crucially, attention deployment is governed by guidance factors (Wolfe & Horowitz, 2017). Among these, *scene structure* plays a dominant role, knowing that a "chimney" is likely on a "roof" enables efficient prioritization. **CVSearch** computationally instantiates this cognitive architecture. Our cognitive *Assess-then-Search* workflow mimics the progression from nonselective to selective attention, while *Visual Complexity* captures scene structure to dynamically prune the search space.

## 3. Preliminary

We consider a standard MLLM framework comprising a vision encoder $\mathcal{V}$, a projector $\mathcal{P}$, and an LLM $\mathcal{F}$. Given an input image $\boldsymbol{I} \in \mathbb{R}^{H \times W \times 3}$ and a text query $\boldsymbol{Q}$, the vision encoder extracts features $\boldsymbol{H}_v$ which are projected into visual tokens $\boldsymbol{Z}_v = \mathcal{P}(\boldsymbol{H}_v)$. Combined with text embeddings $\boldsymbol{Z}_t$, the model generates the response $Y = \{y_1, \ldots, y_S\}$ autoregressively. The probability of generating $Y$ is factorized as:

$$p(y_s \mid I, Q) = \prod_{i=1}^{s-1} \mathcal{F}(y_i \mid y_{<i}, \boldsymbol{Z}_v, \boldsymbol{Z}_t), \qquad (1)$$

The perceptual capability of MLLMs is constrained by the

resolution of $\boldsymbol{Z}_v$ (Tong et al., 2024). Naive resizing to fixed resolutions (e.g., $336^2$) (Liu et al., 2024) causes severe detail loss and distortion. To mitigate this, AnyRes mechanisms (Li et al., 2024a;b) decompose high-resolution (HR) images into flexible grids of local patches alongside a downsampled global view. While AnyRes preserves details, it incurs a prohibitive computational cost, as the sequence length of $\boldsymbol{Z}_v$ scales linearly with the number of patches.

Unlike AnyRes which ingests all patches in a single pass, *visual search* methods iteratively explore local regions to perceive fine-grained details. However, existing approaches face a critical trade-off. *Visual Expert Assisted Search* offers efficiency but is inherently fragile; its success relies entirely on external proposals, leaving the MLLM blind if the expert fails. Conversely, *Scan-based Search* ensure robustness via dense coverage but suffer from computational redundancy and semantic fragmentation due to rigid partitioning.

## 4. Proposed Cognitive Visual Search

### 4.1. Method Overview

Drawing from human cognitive mechanisms that alternate between non-selective and selective attention, **CVSearch** adopts a cognitive *Assess-then-Search* workflow Figure 2(a). The process begins with a global assessment, akin to a human "glimpse." If global information proves insufficient the system triggers *Visual Expert-assisted Search* for rapid localization. In cases the target remain elusive, it transitions to *Scene-aware Scanning* for fine-grained inspection. Crucially, a bidirectional feedback loop integrates these modes, enabling iterative refinement to capture tiny targets.

### 4.2. Visual Expert Assisted Search

**CVSearch** optimizes efficiency by bypassing search when the MLLM's global perception is sufficient. The *Visual Expert* is activated solely when relevant information is elusive, serving as a rapid proposal mechanism to avoid unnecessary scanning costs.

#### 4.2.1. INFORMATION SUFFICIENCY ASSESSMENT

We first quantify the sufficiency of the current visual context. Inspired by (Shen et al., 2025), we use the MLLM's internal confidence in whether the current image $\boldsymbol{I}$ can answer the given query $\boldsymbol{Q}$ as the information sufficiency:

$$c_q(\boldsymbol{I}) = \mathcal{M}(\text{``Yes''} | p_q(\boldsymbol{Q}), \boldsymbol{I}), \qquad (2)$$

where $\mathcal{M}$ represents the MLLM and $p_q(\cdot)$ represents the prompt (*e.g., "Question: {$\boldsymbol{Q}$}. Could you answer the question based on the available visual information? Answer Yes or No."*) used to query the MLLM for calculating the confidence that the answer is "Yes". A higher $c_q(\boldsymbol{I})$ indicates that, for the MLLM, the current image $\boldsymbol{I}$ contains more

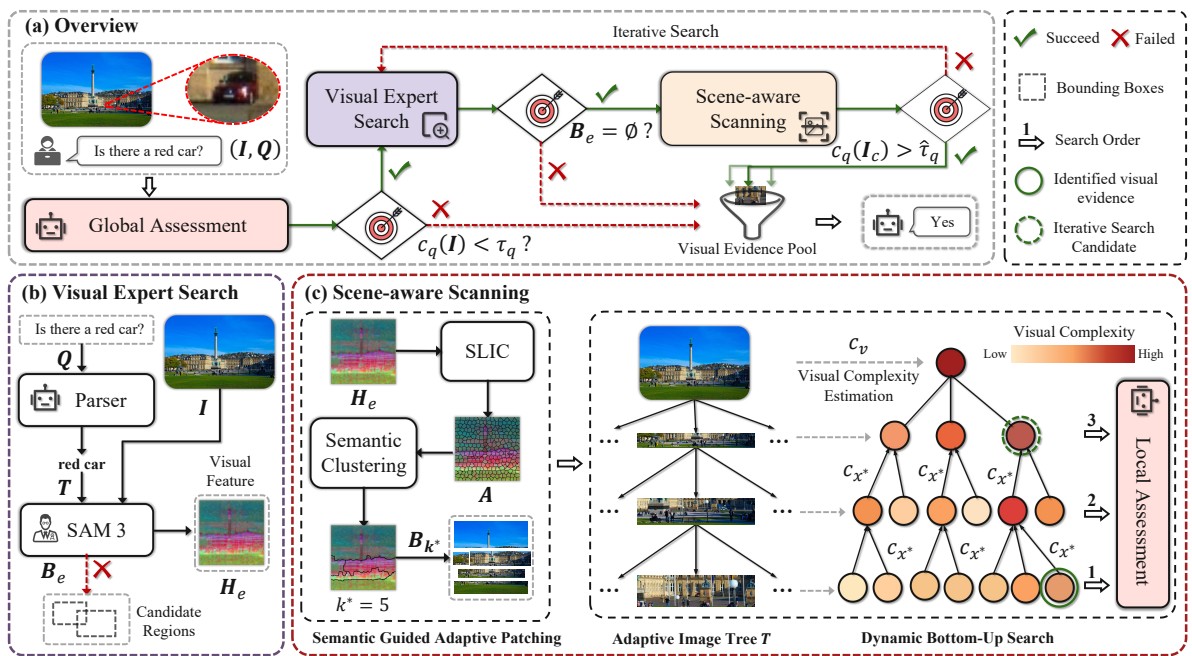

*Figure 2.* Illustration of the **CVSearch** framework. **(a) Workflow.** A cognitive *Assess-then-Search* mechanism triggers **Visual Expert Search** when global information is insufficient ($c_q < \tau_q$). Expert failure (proposals $B_e = \emptyset$) activates **Scene-aware Scanning**, which either yields visual evidence upon success or returns the optimal candidate for iterative search upon failure. **(b) Visual Expert Search.** This module parses queries to prompt a visual expert (SAM 3) for rapid proposals. On failure, extracted visual features are repurposed for the scanning phase. **(c) Scene-aware Scanning.** *Semantic Guided Adaptive Patching* partitions images into semantically coherent regions via adaptive clustering. Subsequently, *Dynamic Bottom-Up Search* prioritizes exploration from leaf nodes and aggregates evidence upwards. If the target remains unconfirmed, the optimal candidate from the first layer guides the next search iteration.

sufficient information to answer the given query $Q$. Consequently, When $c_q(I)$ exceeds the sufficiency threshold $\tau_q$, the global view is deemed adequate for a direct answer response, bypassing fine-grained inspection.

### 4.2.2. VISUAL EXPERT PROPOSAL

For queries unresolved by global image information, we employ a visual expert $\mathcal{E}$ to search for the key objects mentioned in the query. As shown in Figure 2(b), we adopt SAM 3 (Carion et al., 2025) as our visual expert. To enable concept-level prompting, we parse the query $Q$ into a set of target objects $O = \{o_1, o_2, \cdots, o_m\}$. Specifically, we leverage the in-context capability from the LLM base of the MLLM to extract key objects from the query, falling back to SpaCy (Jugran et al., 2021) for noun phrase extraction if necessary. Querying $\mathcal{E}(I, O)$ yields candidate bounding boxes $B_e$ and dense visual features $H_e$.

We further verify whether the proposed regions $B_e$ adequately cover the target objects in $O$. Specifically, this coverage is satisfied when the number of targets (across different categories) segmented by SAM 3 strictly matches the number of extracted target objects. If validated, the system crops $I$ according to $B_e$ for answer generation. Conversely, if the visual expert fails to localize targets, which is a com-

mon failure mode for tiny or abstract objects, **CVSearch** triggers the *Scene-aware Scanning* phase, reusing the pre-extracted $H_e$ to minimize computational overhead.

### 4.3. Scene-aware Scanning

Scan-based visual search methods (Shen et al., 2025; Wang et al., 2025c;d) improve retrieval via fine-grained exploration of local regions, serving as an effective means to compensate for the fragility of vision-expert-assisted search. However, existing scan-based methods partition images into rigid grids and perform top-down tree search without considering scene semantics, leading to substantial search resources being wasted on exploring target-irrelevant regions and correcting initial search errors. To address these issues, we propose *scene-aware scanning*, comprising *Semantic Guided Adaptive Patching* and *Dynamic Bottom-up Search*.

### 4.3.1. SEMANTIC GUIDED ADAPTIVE PATCHING

Standard grid-based partitioning often disrupts semantic integrity by cutting across object boundaries. As shown in Figure 2(c), we utilize the scene structure embedded in the visual feature $H_e$ as a prior for adaptive image patching. We first over-segment $H_e$ into $N$ atomic superpixels $A = \{a_1, a_2, \cdots, a_N\}$ using Simple Linear Iterative Clus-

tering (SLIC) (Achanta et al., 2012) on the feature space and construct a region adjacency graph $G$ to encode connectivity. Subsequently, we partition these atoms into $k$ semantic clusters via Agglomerative Clustering (Müllner, 2011) constrained by $G$. Each semantic cluster consists of spatially adjacent atoms, and the cluster boundary is converted into a region bounding box, which can be used for semantic-preserving image patching.

The number of clusters $k$ significantly impacts the quality of image patching, and the optimal $k$ is scene-dependent. Adhering to the principle that high-quality patching must preserve clear semantic boundaries while minimizing region overlap, we optimize the optimal number of clusters $k^*$ within $[k_{\min}, k_{\max}]$ by minimizing the cost function $\mathcal{L}(k)$:

$$
\begin{aligned}
k^* &= \arg\min_{k \in [k_{\min}, k_{\max}]} \mathcal{L}(k), \\
\mathcal{L}(k) &= \mathcal{L}_o(\boldsymbol{B}_k) - \mathcal{L}_s(\boldsymbol{H}_a, \boldsymbol{l}_k),
\end{aligned}
\tag{3}
$$

where $\mathcal{L}_o$ penalizes spatial overlap among bounding boxes $\boldsymbol{B}_k = \{b_1, b_2, \cdots, b_k\}$ of all clusters, and $\mathcal{L}_s$ is the silhouette score (Vardakas et al., 2024) measuring clustering quality. $\boldsymbol{H}_a = \{\boldsymbol{h}_1, \boldsymbol{h}_2, \cdots, \boldsymbol{h}_N\}$ denotes atomic features derived from the visual feature $\boldsymbol{H}_e$, and $\boldsymbol{l}_k$ are the clustering label corresponding to atomic features. Based on $k^*$, the image is adaptively patched into $k^*$ semantic regions.

### 4.3.2. Dynamic Bottom-Up Search

By recursively applying adaptive patching, we model the HR image $\boldsymbol{I}$ as an adaptive tree $\boldsymbol{T}$ with depth $D$. Each *node* at depth $d$, denoted as $n_{d,t}$, represents a patch view $\boldsymbol{I}_{d,t}$. Furthermore, we introduce a *Visual Complexity* score $c_v$ to quantify the exploration value of an image patch. The $c_v$ of an image patch $\boldsymbol{I}_{d,t}$ is defined as the degree of dispersion of its corresponding atomic features in the semantic direction, quantified by the average cosine similarity:

$$
c_v(\boldsymbol{I}_{d,t}) = \max(0, \ 1 - \frac{1}{|\boldsymbol{R}|} \sum_{i \in \boldsymbol{R}} \text{cosim}(\boldsymbol{h}_i, \bar{\boldsymbol{h}})), \quad (4)
$$

where $\bar{\boldsymbol{h}} = \mathbb{E}_{i \in \boldsymbol{R}}[\boldsymbol{h}_i]$ represents the semantic centroid of the image patch derived from the expectation of atomic features, $\text{cosim}(\cdot, \cdot)$ denotes the cosine similarity, and $\boldsymbol{R}$ represents the set of atomic indices belonging to $\boldsymbol{I}_{d,t}$. A low $c_v$ indicates high feature similarity within the image patch, suggesting that the region is more likely to be a background area with limited semantic information and thus lower exploration value. Conversely, a high $c_v$ reflects low feature similarity in the image patch, indicating that the region is more likely to be a semantically rich foreground area with higher exploration value. To prioritize semantically rich regions, we prune $\boldsymbol{T}$ by discarding nodes with $c_v < \tau_v$.

Traditional hierarchical search methods (Shen et al., 2025; Wang et al., 2025c;d) traverses trees top-down (root-to-leaf).

However, at the initial stage, models struggle to accurately perceive small objects, often resulting in erroneous search paths. As shown in Figure 2(c), we propose a *Dynamic Bottom-Up Search* mechanism to robustly identify target objects. The search initiates at the deepest layer of nodes, which provide detailed local information to enhance the MLLM's precision in perceiving small targets. If unsuccessful at this layer, the collected search information is aggregated and propagated to the layer containing the parent nodes, where the search continues.

Within each layer, nodes are sorted by their priority values $c_x$ to determine the order of node visitation. For each node $n_{d,t}$, we adopt the MLLM to assess the existence confidence $c_o$ of the target objects in $\boldsymbol{O}$. $c_o$ is calculated by Equation (2) with prompt $p_o(o_i)$ (*e.g.*, "*Is there a $\{o_i\}$ in the image? Answer Yes or No.*") and the image patch $\boldsymbol{I}_{d,t}$. The priority value for a node is the weighted sum of the *Visual Complexity* score $c_v$, the existence confidence $c_o$, and the priority value $c_x^*$ aggregated from child nodes:

$$
c_x = \alpha \cdot c_v + \beta \cdot c_o + \gamma \cdot c_x^*, \tag{5}
$$

where $\alpha$, $\beta$, and $\gamma$ are hyperparameter that balance the effects of different terms. For a node with child nodes, the value of $c_x^*$ is the maximum priority value among its children, whereas for a node without child nodes, $c_x^* = 0$. The information sufficiency $c_q$ of ranked nodes is calculated using Equation (2) to serve as the stopping criterion for the bottom-up search. For multi-target scenarios, we adopt the same setting as ZoomEye (Shen et al., 2025) and construct a decoupled query $Q_d$ (*e.g.*, "*What is the appearance of the $\{o_i\}$*") for each target. For the search termination criterion, we adopt an adaptive sufficiency threshold $\tau_{curr}$ instead of a static stopping rule. The search begins with a rigorous standard ($\tau_{curr} = \tau_q$) to guarantee that easy samples are resolved with high certainty. For more challenging scenarios where the model may hesitate, $\tau_{curr}$ gradually decays to permit valid but less confident predictions, ensuring the retrieval of subtle details. Crucially, this relaxation is bounded by a minimum threshold $\hat{\tau}_q$, which serves as a safeguard to reject regions that lack sufficient semantic evidence for reasoning. The search terminates once $c_q$ exceeds a predefined threshold $\tau_{curr}$. If the search proceeds to completion at depth $d = 1$ without meeting this condition, the top-ranked node from that layer is fed back to the visual expert as the candidate region for iterative search.

## 5. Experiments

### 5.1. Experimental Setup

**Datasets**. We evaluate *CVSearch* on a comprehensive suite of high-resolution benchmarks. For **HR-specific evaluation**, we use *V\* Bench* (Wu & Xie, 2024) (avg. res. 2246×1582), which targets attribute recognition and spatial reasoning,

*Table 1.* Performance comparison of our **CVSearch** (integrated with several advances models) with existing works on high-resolution benchmarks. **FSP**: Fine-grained Single-instance Perception; **FCP**: Finegrained Cross-instance Perception.

| Method | V* Bench | | | HR-Bench 4K | | | HR-Bench 8K | | |
|---|---|---|---|---|---|---|---|---|---|
| | *Attribute* | *Spatial* | *Overall* | *FSP* | *FCP* | *Overall* | *FSP* | *FCP* | *Overall* |
| *Closed-source MLLMs* | | | | | | | | | |
| GPT 4o (Hurst et al., 2024) | - | - | 66.0 | 70.0 | 48.0 | 59.0 | 62.0 | 49.0 | 55.5 |
| QWen-VL-max (Bai et al., 2023) | - | - | - | 65.0 | 52.0 | 58.5 | 54.0 | 51.0 | 52.5 |
| *Open-source MLLMs* | | | | | | | | | |
| LLaVA-v1.6-7B (Li et al., 2024b) | 60.9 | 63.2 | 61.8 | 49.0 | 46.8 | 47.9 | 37.3 | 44.3 | 40.8 |
| LLaVA-v1.6-13B (Li et al., 2024b) | 60.0 | 64.5 | 61.8 | 49.8 | 41.3 | 45.5 | 38.0 | 38.3 | 38.1 |
| LLaVA-v1.6-34B (Li et al., 2024b) | - | - | - | 55.3 | 50.5 | 52.9 | 44.5 | 50.3 | 47.4 |
| LLaVA-HR-X-7B (Luo et al., 2025) | 51.3 | 64.5 | 56.5 | 57.8 | 46.3 | 52.0 | 42.0 | 41.3 | 41.6 |
| LLaVA-HR-X-13B (Luo et al., 2025) | - | - | - | 61.3 | 46.0 | 53.6 | 49.5 | 44.3 | 46.9 |
| Qwen2.5-VL-32B (Bai et al., 2025) | 83.5 | 89.5 | 85.9 | 89.3 | 60.3 | 74.8 | 86.5 | 56.8 | 71.6 |
| Yi-VL-34B (Young et al., 2024) | - | - | - | 46.0 | 42.8 | 44.4 | 39.5 | 38.5 | 39.0 |
| InternVL3-38B (Zhu et al., 2025) | 77.4 | 77.6 | 77.5 | 83.5 | **69.0** | 76.3 | 71.3 | 62.8 | 67.0 |
| *Baseline and CVSearch* | | | | | | | | | |
| LLaVA-OV-7B (Li et al., 2024a) | 75.7 | 75.0 | 75.4 | 72.0 | 54.0 | 63.0 | 67.3 | 52.3 | 59.8 |
| *-w/ CVSearch* | **95.7** | 85.5 | **91.6** | 89.5 | 61.8 | 75.6 | 89.0 | 60.5 | 74.8 |
| Qwen2.5-VL-7B (Bai et al., 2025) | 73.9 | 67.1 | 71.2 | 85.2 | 52.2 | 68.8 | 78.8 | 51.8 | 65.3 |
| *-w/ CVSearch* | 93.0 | 85.5 | 90.1 | 91.5 | 61.8 | 76.6 | 90.0 | 61.3 | 75.6 |
| InternVL2.5-8B (Chen et al., 2024) | 67.8 | 71.1 | 69.1 | 75.8 | 56.3 | 66.0 | 61.5 | 53.3 | 57.4 |
| *-w/ CVSearch* | 86.1 | **93.4** | 89.0 | **93.0** | 61.0 | **77.0** | 92.3 | **63.0** | **77.6** |

and **HR-Bench** (Wang et al., 2025c), comprising **8K** and **4K** subsets for fine-grained single- and cross-instance perception. *HR-Bench 4K* is derived by cropping relevant regions from the 8K-resolution images in *HR-Bench 8K*. For **General and Real-world evaluation**, we employ **MME-RealWorld-Lite** (Zhang et al., 2024b) and **TreeBench** (Wang et al., 2025a). These manually curated benchmarks feature diverse real-world subtasks with high average resolutions ($\sim 2000 \times 1500$). Additionally, we test on **FineRS-4K** (Zhang et al., 2025b), a specialized UAV-captured high-resolution dataset for the perception and reasoning of ultra-small objects.

**Implementation Details**. We select Qwen2.5-VL-7B (Bai et al., 2025), LLaVA-OV (OneVision)-7B (Li et al., 2024a), and InternVL2.5-8B (Chen et al., 2024) as baseline MLLMs. The information sufficiency threshold $\tau_q$ is set to 0.9, the search termination threshold $\hat{\tau}_q$ to 0.5, and the tree pruning threshold $\tau_v$ to 0.4. The minimum and maximum numbers of clusters, $k_{\min}$ and $k_{\max}$, are set to 4 and 8, respectively. The depth $D$ of the adaptive image tree $T$ is set to 2 for single-object cases ($m = 1$) and 3 for multi-object cases ($m > 1$). The hyperparameters $\alpha$, $\beta$, and $\gamma$ are set to 0.2, 0.4, and 0.4, respectively. All experiments are conducted on four NVIDIA A6000 GPUs, while inference speed evaluation is performed using a single GPU.

### 5.2. Main Experimental Results

**Results on HR Benchmark**. As shown in Table 1, integrating our **CVSearch** framework with various open-source MLLMs consistently yields substantial performance gains across all high-resolution benchmarks, underscoring its robust model-agnostic effectiveness. Our method significantly enhances both *FSP* and *FCP* tasks, achieving state-of-the-art results on *HR-Bench 4K* and *HR-Bench 8K*. Notably, when applied to InternVL2.5-8B, **CVSearch** boosts overall accuracy by +20.2% on *HR-Bench 8K* and by +11.0 on *HR-Bench 4K*. On *V* Bench*, **CVSearch** dramatically boosts performance as LLaVA-OV-7B improves from 75.4 to 91.6 and Qwen2.5-VL-7B reaches 90.1, rivaling much larger closed-source models. These results demonstrate that our method effectively empowers existing MLLMs with the capability to perceive high-resolution images more accurately.

**Compared with Visual Search Methods**. We compare our proposed **CVSearch** against representative visual search paradigms, comprising expert-assisted methods (SEAL (Wu & Xie, 2024), DyFo (Li et al., 2025a)) and scan-based approaches (Zoom Eye (Shen et al., 2025), RAP (Wang et al., 2025d)). As shown in Table 2, our method consistently outperforms all baselines across all evaluated benchmarks. Specifically, while scanning-based methods like Zoom Eye and RAP offer improvements over the vanilla LLaVA-OV-7B, they still fall short of **CVSearch**, which achieves 91.6

*Table 2.* Performance comparison between our **CVSearch** and other search-based method.

| Method | V* | HR-4K | HR-8K |
|---|---|---|---|
| SEAL | 75.4 | - | - |
| DyFo | 81.2 | - | - |
| LLaVA-OV-7B | 75.4 | 63.0 | 59.8 |
| *-w/ Zoom Eye* | 90.6 | 69.6 | 69.3 |
| *-w/ RAP* | 79.6 | 71.0 | 67.6 |
| ***-w/ CVSearch*** | **91.6** | 75.6 | 74.8 |
| Qwen2.5-VL-7B | 71.2 | 68.8 | 65.3 |
| *-w/ Zoom Eye* | 85.3 | 72.5 | 69.8 |
| *-w/ RAP* | 84.8 | 74.8 | 76.0 |
| ***-w/ CVSearch*** | 90.1 | 76.6 | 75.6 |
| InternVL2.5-8B | 69.1 | 66.0 | 57.4 |
| *-w/ Zoom Eye* | 84.8 | 75.1 | 73.6 |
| *-w/ RAP* | 88.5 | 76.0 | 74.0 |
| ***-w/ CVSearch*** | 89.0 | **77.0** | **77.6** |

*Table 3.* Comparison of our **CVSearch** against the baseline MLLM on three general multimodal benchmarks. **MME-RW-L** stands for MME-RealWorld-Lite.

| Method | MME-RW-L | TreeBench | FineRS-4K |
|---|---|---|---|
| LLaVA-OV-7B | 43.7 | 37.3 | 72.0/49.7 |
| *-w/ CVSearch* | **48.8** | 38.8 | 77.4/**61.2** |
| Qwen2.5-VL-7B | 42.3 | 37.0 | 80.4/59.1 |
| *-w/ CVSearch* | 46.7 | **40.7** | **82.5**/58.3 |
| InternVL2.5-8B | 44.9 | 25.7 | 71.6/51.6 |
| *-w/ CVSearch* | 46.1 | 29.1 | 78.5/57.9 |

on *V* Bench*, surpassing the strong expert-assisted DyFo (81.2) by a wide margin. On high-resolution benchmarks, our method proves highly scalable, achieving state-of-the-art scores of 77.0 on *HR-Bench 4K* and 77.6 on *HR-Bench 8K* with InternVL2.5-8B. These results highlight the clear advantage of our cognitive search mechanism compared to fixed scanning strategies or external expert dependency. More comparison results with other HR image processing methods can be found in the Appendix B.1.

**Results on General Benchmarks**. To assess the generalization capability of **CVSearch** in complex real-world scenarios, we evaluate it on three challenging multimodal benchmarks (Table 3). Note that for FineRS-4K, we provide accuracy metrics for both multiple-choice and open-ended visual question answering (MVQA/OVQA). As we can see, **CVSearch** yields consistent improvements across all baselines, demonstrating its robustness in handling complex visual reasoning tasks. Specifically, on *MME-RealWorld-Lite* and *TreeBench*, our method improves the reasoning accuracy of Qwen2.5-VL-7B by +4.4 and +3.7, respectively. A more pronounced advantage is observed on *FineRS-4K*, where identifying tiny objects is critical. **CVSearch** en-

*Table 4.* Comparison of our **CVSearch** against the baseline MLLM with varying parameter sizes and generations.

| Method | V* | HR-4K | HR-8K |
|---|---|---|---|
| Qwen2.5-VL-3B | 77.0 | 65.9 | 62.9 |
| ***-w/ CVSearch*** | 91.1 | 70.5 | 67.3 |
| Qwen3-VL-2B | 79.1 | 71.3 | 67.8 |
| ***-w/ CVSearch*** | 92.2 | 74.0 | 73.6 |
| Qwen3-VL-4B | 89.5 | 76.3 | 71.6 |
| ***-w/ CVSearch*** | **93.7** | 77.4 | 75.1 |
| Qwen3-VL-8B | 88.0 | 79.1 | 74.5 |
| ***-w/ CVSearch*** | 91.6 | 79.5 | 76.5 |
| Qwen3-VL-32B | 86.9 | 76.5 | 70.9 |
| ***-w/ CVSearch*** | 89.5 | **80.3** | **78.4** |

ables LLaVA-OV-7B to surge from 49.7 to 61.2 (+11.5) in the challenging OVQA setting, while also boosting InternVL2.5-8B to 78.5 in MVQA. These results confirm that the cognitive search mechanism of **CVSearch** effectively extends to complex, open-ended real-world scenarios.

### 5.3. Model Analysis

**Scalability across Sizes and Generations**. To investigate the scalability of our framework, we evaluate **CVSearch** on models with varying parameter sizes (ranging from 2B to 32B) and generations (Qwen2.5-VL vs. the latest Qwen3-VL). As shown in Table 4, our method consistently enhances performance across all model variants. For smaller models, the improvements are particularly substantial. For instance, on the *V* Bench*, **CVSearch** boosts Qwen2.5-VL-3B by +14.1 and Qwen3-VL-2B by +13.1. Crucially, to address potential concerns regarding diminishing returns on more advanced backbones, we scale our evaluation up to Qwen3-VL-32B. Despite the powerful native perception capabilities of this large model, **CVSearch** continues to provide substantial gains, boosting performance by +7.5 on HR-8K, +3.8 on HR-4K, and +2.6 on V*. This validates that our search mechanism effectively resolves high-resolution bottlenecks even for highly advanced MLLMs, serving as a scalable solution that benefits both resource-constrained deployment (on small models) and high-performance computing (on large models).

**Impact of Tree Depth**. We investigate the influence of the search depth $D$ on perception accuracy, as illustrated in Figure 3. Given that *V* Bench* comprises single-object Attribute tasks ($m = 1$) and multi-object Spatial tasks ($m = 2$), we observe a clear correlation between target count and optimal depth. Specifically, Attribute tasks perform exceptionally well at depth 2, whereas Spatial tasks peak at depth 3. Furthermore, under current benchmark resolutions (up to 8K), extending the search to depth 4 offers negligible accuracy improvement but may incur penalties in

*Table 5.* Search efficiency comparison with expert-based and scan-based methods. We measure search efficiency by *Throughput* (samples per minute) on a single GPU using Qwen2.5-VL-7B. The *Accuracy* and *Throughput* are represented as *Acc* and *Thr*, respectively.

| Visual Search Method | V* Bench | | HR-Bench 4K | | HR-Bench 8K | |
|---|---|---|---|---|---|---|
| | Acc ↑ | Thr ↑ | Acc ↑ | Thr ↑ | Acc ↑ | Thr ↑ |
| Qwen2.5-VL-7B | 71.2 | 8.30 | 68.8 | 7.62 | 65.3 | 7.62 |
| *-w/ Visual Expert (SAM 3)* | 84.3 | 3.60 | 71.9 | 5.59 | 68.1 | 5.30 |
| *-w/ Scan-based (Zoom Eye)* | 85.3 | 0.68 | 72.5 | 1.29 | 69.8 | 0.68 |
| *-w/ Scan-based (RAP)* | 84.8 | 0.66 | 74.8 | 1.22 | 76.0 | 0.58 |
| ***-w/ CVSearch*** | 90.1 | 1.02 | 76.6 | 3.77 | 75.6 | 1.92 |

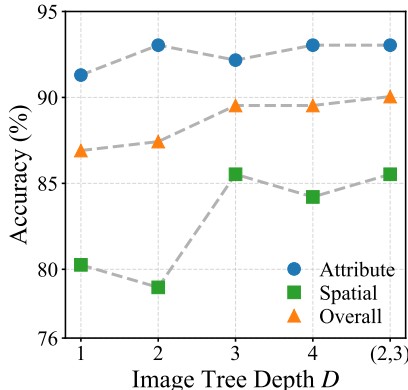

*Figure 3.* Impact of Image Tree Depth $D$ on *V* Bench* performance. Evaluated with Qwen2.5-VL-7B, we compare fixed depths ($D = 1$ to 4) against our adaptive strategy, denoted as $(2, 3)$.

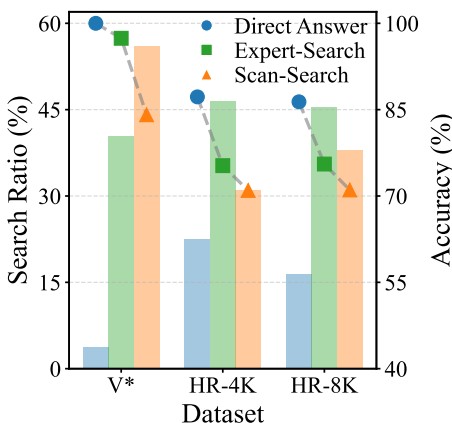

*Figure 4.* Performance analysis of different search modes. The bar chart (left axis) displays the usage frequency of each mode, while the scatter plot (right axis) reports the corresponding accuracy.

both semantic integrity due to fragmentation and computational overhead due to increased search time. However, it is important to note that the optimal tree depth is inherently resolution-dependent. For ultra-high-resolution imagery (e.g., 16K and beyond), the massive search space renders $D = 3$ too coarse, making deeper search depths ($D \geq 4$) essential for capturing microscopic details. Crucially, extending to deeper levels does not lead to a cost explosion; our SGAP and Branch Pruning mechanisms aggressively filter irrelevant background nodes prior to MLLM inference, concentrating compute exclusively on high-value regions to balance ultra-fine extraction with practical efficiency. For our primary evaluation settings, rather than relying on a fixed global hyperparameter, we employ a rule-based dynamic routing strategy at the sample level. By explicitly setting the search depth to $D = 2$ for single objects and $D = 3$ for multiple objects, this target-conditioned design effectively navigates the trade-off to achieve superior accuracy without compromising efficiency.

**Visual Search Efficiency**. To comprehensively evaluate the computational overhead, Table 5 presents a consolidated analysis of inference throughput (samples per minute). While *CVSearch* naturally introduces inference overhead compared to the vanilla model, it effectively overcomes the inherent limitations of both expert-assisted and scan-based

paradigms. Compared to the lightweight expert-assisted approach (SAM 3), *CVSearch* delivers substantial accuracy improvements (e.g., +4.7% on HR-4K) while maintaining competitive throughput. More importantly, rather than making a speed-accuracy trade-off, our method strictly outperforms existing scan-based frameworks in both dimensions. It achieves significantly higher accuracy while being substantially faster. For instance, on the *HR-Bench 4K*, *CVSearch* achieves a throughput of 3.77, which is nearly 3× faster than Zoom Eye and RAP. This dual advantage in both efficiency and performance is directly attributable to our adaptive *Cognitive Assess-then-Search* workflow, which intelligently bypasses exhaustive scanning for samples with sufficient global information and restricts iterative search strictly to highly informative regions.

**Performance of Different Search Modes**. Figure 4 analyzes the activation frequency and accuracy of each search mode within our *Assess-then-Search* workflow. We observe a distinct alignment between the selected mode and sample difficulty. The consistently high accuracy of **Direct Answer** and **Expert-Search** confirms that our mechanism effectively filters and resolves easy or salient samples, avoiding unnecessary computational costs. The **Scan-Search** focuses solely on hard examples that failed in earlier stages, resulting in lower accuracy.

*Table 6.* Ablation study on our **CVSearch** components. We measure search efficiency, denoted as *Thr*, by *Throughput* (samples per minute) on a single GPU. The *Accuracy* is denoted as *Acc*. *SGAP* represents our proposed *Semantic Guided Adaptive Patching*.

| Method | V* Bench | | HR-Bench 4K | | HR-Bench 8K | |
|---|---|---|---|---|---|---|
| | Acc ↑ | Thr ↑ | Acc ↑ | Thr ↑ | Acc ↑ | Thr ↑ |
| Qwen2.5-VL-7B | 71.2 | 8.30 | 68.8 | 7.62 | 65.3 | 7.62 |
| *-w/ SAM 3* | 84.3 | 3.60 | 71.9 | 5.59 | 68.1 | 5.30 |
| *-w/ SAM 3 + Rigid Grid + Top-down search* | 84.8 | 0.68 | 73.5 | 2.02 | 70.3 | 1.15 |
| *-w/ SAM 3 + SGAP + Top-down search* | 84.8 | 0.68 | 72.3 | 2.14 | 70.8 | 1.12 |
| *-w/ SAM 3 + SGAP + Bottom-up search* | 86.4 | 1.29 | 76.8 | 3.81 | 74.9 | 2.18 |
| *-w/ CVSearch* | 90.1 | 1.02 | 76.6 | 3.77 | 75.6 | 1.92 |

**Effectiveness of Key Components**. We perform a progressive ablation study in Table 6 to validate the contribution of each module in **CVSearch**. First, incorporating SAM 3 improves the baseline *V* Bench* accuracy by +13.1, establishing a strong foundation. Second, comparing scanning strategies, our proposed **SGAP + Bottom-up search** significantly outperforms the traditional "Rigid Grid + Top-down" baseline. It not only boosts *HR-Bench 4K* accuracy from 73.5 to 76.8 but also drastically reduces inference latency. This highlights the effectiveness of our semantic-aware patching and complexity-driven exploration in eliminating redundant calculations. Finally, the full **CVSearch** framework achieves the best overall performance across all benchmarks, validating the synergy of our cognitive architecture.

Ablation studies on the information sufficiency threshold $\tau_q$ are provided in Appendix B.2. Furthermore, Appendix C.1 demonstrates how our Semantic Guided Adaptive Patching strategy adaptively partitions images based on visual complexity. Subsequently, Appendix C.2 provides a detailed analysis of representative successful inference examples as well as typical failure cases to discuss the current capabilities and limitations of our approach.

## 6. Conclusions

In this paper, we present **CVSearch**, a novel training-free framework that resolves the efficiency-robustness dilemma in high-resolution visual perception for MLLMs. By reformulating visual search as an cognitive, hierarchical decision-making process, **CVSearch** effectively balances the computational overhead of exhaustive scanning with the reliability risks of expert dependency. Our core contributions, *Semantic-Guided Adaptive Patching Dynamic Bottom-Up Search*, facilitate semantic-preserving partitioning and robust iterative exploration. Extensive experiments validate that **CVSearch** achieves superior performance across diverse benchmarks while maintaining high inference efficiency. We hope this work inspires further research into cognitive visual strategies for fine-grained visual understanding.

## Limitations

While **CVSearch** demonstrates significant improvements in high-resolution visual search, we acknowledge two primary limitations. First, regarding computational efficiency, although our method is substantially faster than existing scan-based frameworks, the iterative search process inevitably makes it slower than the vanilla single-pass MLLM and the lightweight expert-assisted baselines. To further mitigate this latency, future work can focus on system-level acceleration, such as evaluating independent nodes at the same search depth in parallel via batched inference and integrating memory-efficient mechanisms like FlashAttention into the base model. Second, regarding the visual expert, our current experiments rely exclusively on SAM 3 to generate region proposals. Consequently, the generalization capability of the framework when integrated with different types of visual experts remains unexplored. Future research will investigate substituting or combining diverse visual foundation models to evaluate and enhance the system's robustness and adaptability across various domains and task requirements.

## Acknowledgements

We sincerely thank the anonymous reviewers and chairs for their efforts and constructive suggestions, which have greatly helped us improve the manuscript. This work is supported in part by the National Natural Science Foundation of China under grants 62536003, 62521006, 624B2088 and 62576122, and by the project of Peng Cheng Laboratory (PCL2025A14).

## Impact Statement

Our work introduces **CVSearch**, a training-free framework that optimizes the trade-off between visual search efficiency and accuracy for multimodal large language models (MLLMs). Specifically, via an *Assess-then-Search* workflow, **CVSearch** adaptively prioritizes rapid inference modes by relying on global context or expert proposals, and only resorts to extensive scanning when necessary. Furthermore, by incorporating *Semantic Guided Adaptive Patching* and

*Dynamic Bottom-Up Search*, the framework leverages scene semantics to significantly accelerate the scan-based exploration, enabling MLLMs to capture fine-grained details with minimal computational redundancy. We believe that by prioritizing adaptive resource allocation and the mitigation of computational redundancy, this work steers the field toward more efficient, scalable, and accessible high-resolution multimodal understanding.

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

# Appendix

In this appendix, we provide comprehensive implementation details, extended experimental results, and case analyses to further support the contributions of *CVSearch*. The appendix is organized as follows:

- **Appendix A**: Details the algorithmic implementation of our cognitive framework, including the algorithm workflow for the overall inference process, the *Semantic Guided Adaptive Patching* mechanism, and the *Dynamic Bottom-Up Search* strategy.

- **Appendix B**: Presents additional quantitative evaluations, including comparisons with a broader range of high-resolution image perception methods and an in-depth ablation study on the information sufficiency threshold $\tau_q$, which governs the **cognitive gating** process.

- **Appendix C**: Provides qualitative analysis through extensive case studies and visualizations, demonstrating the effectiveness of our cognitive search process in complex scenarios.

## A. Algorithm Implementation Details

In this section, we provide the formal algorithms and detailed implementation steps for the key components that constitute the **cognitive architecture** of *CVSearch*.

### A.1. Overall Cognitive Framework of *CVSearch*

The cognitive inference process is outlined in Algorithm 1. Mirroring human cognitive control, the system first performs a **Global Cognitive Assessment** to evaluate information sufficiency ($c_q > \tau_q$). If the global gist is insufficient, it triggers the Visual Expert ($\mathcal{E}$) to propose target regions. Upon expert failure, the method switches to Scene-aware Scanning (using Algorithm 2 and Algorithm 3). Crucially, a feedback loop enables iterative refinement, simulating the human "re-check" mechanism. In case where the bottom-up search concludes at depth 1 without meeting the sufficiency criteria, the top-ranked node from this layer is cropped and re-fed into the pipeline for further cognitive inspection.

### A.2. Implementation of Semantic Guided Adaptive Patching

Algorithm 2 details the construction of the adaptive image tree $\boldsymbol{T}$. This process instantiates the system's structural understanding of the scene. We initialize atomic superpixels using SLIC and build a Region Adjacency Graph. The optimal number of clusters $k^*$ is determined by minimizing the cost function $\mathcal{L}(k)$ (Eq. 3). Finally, we apply visual complexity pruning, retaining only nodes where $c_v$ exceeds the threshold $\tau_v$ to filter out non-informative background regions.

### A.3. Implementation of Dynamic Bottom-Up Search

Algorithm 3 describes the search strategy traversing from depth $D$ to 1. In each layer, nodes are sorted by priority $c_x$ (Eq. 5). To handle uncertainty in small object perception, To handle uncertainty in small object perception, we employ a dynamic termination threshold $\tau_{curr}$ that decays from $\tau_q$ to a lower bound $\hat{\tau}_q$ as the search step increases. The algorithm returns FOUND if a definitive answer is reached, or the top-ranked node from the first layer for fallback refinement in the main framework.

---

**Algorithm 1** The Cognitive Inference Process of *CVSearch*

---

**Require:** Image $I$, Query $Q$, MLLM $\mathcal{M}$, Visual Expert $\mathcal{E}$ (SAM 3)
**Require:** Thresholds: $\tau_q$ (Sufficiency), $\hat{\tau}_q$ (Lower Bound), $\Delta\tau$ (Decay Step)
**Ensure:** Response $R$
1: **Initialize:** Current context $I_{curr} \leftarrow I$, Iteration $iter \leftarrow 0$
2: **while** $iter < MAX\_ITER$ **do**
3:     **Phase 1: Global Cognitive Assessment**
4:     Calculate sufficiency score $c_q(I_{curr})$ using Eq. 2.
5:     **if** $c_q(I_{curr}) > \tau_q$ **then**
6:       $R \leftarrow \mathcal{M}(I_{curr}, Q)$ {Global information is sufficient}
7:       **return** $R$
8:     **end if**
9:     **Phase 2: Visual Expert-assisted Search**
10:     Parse target objects $\boldsymbol{O} = \{o_1, \ldots, o_m\}$ from $Q$.
11:     Generate proposals $\boldsymbol{B}_e$ and features $\boldsymbol{H}_e$ via $\mathcal{E}(I_{curr}, \boldsymbol{O})$.
12:     **if** Verification($\boldsymbol{B}_e, \boldsymbol{O}$) is Successful **then**
13:       $I_{crop} \leftarrow \text{Crop}(I_{curr}, \boldsymbol{B}_e)$
14:       $R \leftarrow \mathcal{M}(I_{crop}, Q)$
15:       **return** $R$
16:     **else**
17:       **Phase 3: Scene-aware Scanning (SGAP & Dynamic Bottom-Up Search)**
18:       Construct Adaptive Tree $\boldsymbol{T}$ using **Algorithm 2** with $\boldsymbol{H}_e$.
19:       {Search for the best local region}
20:       $Status, Node \leftarrow$ **Algorithm 3** $(\boldsymbol{T}, \boldsymbol{O}, Q, \tau_q, \hat{\tau}_q, \Delta\tau)$
21:       **if** $Status ==$ FOUND **then**
22:         {Search successful: Answer based on the found region}
23:         $I_{final} \leftarrow \text{Crop}(I_{curr}, Node)$
24:         $R \leftarrow \mathcal{M}(I_{final}, Q)$
25:         **return** $R$
26:       **else**
27:         {Search inconclusive: Use best candidate for iterative refinement}
28:         $I_{curr} \leftarrow \text{Crop}(I_{curr}, Node)$
29:         $iter \leftarrow iter + 1$
30:         **if** Region too small **then**
31:           **break**{Stop if region is atomic}
32:         **end if**
33:       **end if**
34:     **end if**
35: **end while**
36: **return** $\mathcal{M}(I_{curr}, Q)$ {Fallback answer}

---

---

**Algorithm 2** Semantic Guided Adaptive Patching

---

**Require:** Visual Features $\boldsymbol{H}_e$, Clustering Range $[k_{\min}, k_{\max}]$
**Ensure:** Adaptive Image Tree $\boldsymbol{T}$
 1: **function** BUILDTREE($\boldsymbol{H}, d$)
 2:   **Step 1: Atom Generation**
 3:   Generate $N$ atomic superpixels $\boldsymbol{A}$ using SLIC on $\boldsymbol{H}$.
 4:   Construct Region Adjacency Graph $G$ on $\boldsymbol{A}$.
 5:   **Step 2: Adaptive Clustering Optimization**
 6:   $k^* \leftarrow k_{\min}, \mathcal{L}_{min} \leftarrow \infty$
 7:   **for** $k = k_{\min}$ to $k_{\max}$ **do**
 8:     Partition $\boldsymbol{A}$ into $k$ clusters using Agglomerative Clustering on $G$.
 9:     Calculate cost $\mathcal{L}(k) = \mathcal{L}_o(\boldsymbol{B}_k) - \mathcal{L}_s(\boldsymbol{H}_a, \boldsymbol{l}_k)$ (Eq. 3).
10:     **if** $\mathcal{L}(k) < \mathcal{L}_{min}$ **then**
11:       $\mathcal{L}_{min} \leftarrow \mathcal{L}(k), k^* \leftarrow k$
12:     **end if**
13:   **end for**
14:   **Step 3: Node Construction & Pruning**
15:   Partition current region into $k^*$ patches based on optimal clusters.
16:   **for** each patch view $\boldsymbol{I}_{patch}$ **do**
17:     Calculate Visual Complexity $c_v(\boldsymbol{I}_{patch})$ using Eq. 4.
18:     **if** $c_v(\boldsymbol{I}_{patch}) \geq \tau_v$ **then**
19:       Add $\boldsymbol{I}_{patch}$ as a node to $\boldsymbol{T}$.
20:       Extract features $\boldsymbol{H}_{sub}$ for $\boldsymbol{I}_{patch}$.
21:       BUILDTREE($\boldsymbol{H}_{sub}, d + 1$) {Recursive construction}
22:     **end if**
23:   **end for**
24:   **end function**
25: **Initialize** root node with global feature $\boldsymbol{H}_e$.
26: BUILDTREE($\boldsymbol{H}_e, 0$)
27: **return** $\boldsymbol{T}$

---

---

**Algorithm 3** Dynamic Bottom-Up Search Strategy

---

**Require:** Image Tree $T$, Targets $O$, Query $Q$
**Require:** Initial Threshold $\tau_q$, Lower Bound $\hat{\tau}_q$, Decay Step $\Delta\tau$
**Ensure:** Status, Candidate Node
 1: **Initialize:** Step counter $k_{step} \leftarrow 0$
 2: **for** depth $d = D$ down to 1 **do**
 3:      **Step 1: Layer-wise Priority Calculation**
 4:      Get nodes $N_d$ in current layer $d$.
 5:      **for** each node $n \in N_d$ **do**
 6:          Calculate existence confidence $c_o$ for $O$ using $\mathcal{M}$.
 7:          $c_x^* \leftarrow \max_{child \in Children(n)} c_x(child)$ (set 0 if leaf).
 8:          Compute Priority $c_x = \alpha c_v + \beta c_o + \gamma c_x^*$ (Eq. 4).
 9:      **end for**
10:      Sort $N_d$ by $c_x$ descending.
11:      **Step 2: Dynamic Search within Layer**
12:      **for** each node $n \in N_d$ **do**
13:          Calculate sufficiency $c_q(n)$ for $Q$ (or decoupled $Q_d$).
14:          {Calculate dynamic threshold bounded by $\hat{\tau}_q$}
15:          $\tau_{curr} \leftarrow \max(\tau_q - k_{step} \cdot \Delta\tau, \ \hat{\tau}_q)$
16:          **if** $c_q(n) > \tau_{curr}$ **then**
17:              **return** FOUND, $n$ {Target found, return for answering}
18:          **else**
19:              $k_{step} \leftarrow k_{step} + 1$ {Relax criteria for next attempt}
20:          **end if**
21:      **end for**
22:      {If not found in this layer, proceed to the parent layer $(d-1)$}
23: **end for**
24: **Step 3: Feedback Generation**
25: {Search exhausted without meeting sufficiency criteria}
26: Get top-ranked node $n_{top}$ from Depth 1 (Root's children).
27: **return** NOT_FOUND, $n_{top}$ {Return best guess for iterative refinement}

---

*Table 7.* Performance comparison of our *CVSearch* and more HR image processing methods.

| Method | V* Bench | | | HR-Bench 4K | | | HR-Bench 8K | | |
|---|---|---|---|---|---|---|---|---|---|
| | *Attribute* | *Spatial* | *Overall* | *FSP* | *FCP* | *Overall* | *FSP* | *FCP* | *Overall* |
| VisCrop (Zhang et al., 2025a) | - | - | 62.3 | - | - | 46.3 | - | - | 35.8 |
| DC$^2$ (Wang et al., 2025c) | 49.6 | 59.2 | 51.6 | 45.3 | 37.0 | 41.1 | 36.5 | 33.3 | 34.9 |
| Pixel-Reasoner (Wang et al., 2025b) | 83.5 | 76.3 | 80.6 | 86.0 | 60.3 | 72.9 | 80.0 | 54.3 | 66.9 |
| DeepEyes (Zheng et al., 2025) | 91.3 | 88.2 | 90.1 | 91.3 | 59.0 | 75.1 | 86.8 | 58.5 | 72.6 |
| LLaVA-OV-7B (Li et al., 2024a) | 75.7 | 75.0 | 75.4 | 72.0 | 54.0 | 63.0 | 67.3 | 52.3 | 59.8 |
| *-w/CVSearch* | **95.7** | 85.5 | **91.6** | 89.5 | **61.8** | 75.6 | 89.0 | 60.5 | 74.8 |
| Qwen2.5-VL-7B (Bai et al., 2025) | 73.9 | 67.1 | 71.2 | 85.2 | 52.2 | 68.8 | 78.8 | 51.8 | 65.3 |
| *-w/CVSearch* | 93.0 | 85.5 | 90.1 | 91.5 | **61.8** | 76.8 | 90.0 | 61.3 | 75.6 |
| InternVL2.5-8B (Chen et al., 2024) | 67.8 | 71.1 | 69.1 | 75.8 | 56.3 | 66.0 | 61.5 | 53.3 | 57.4 |
| *-w/CVSearch* | 86.1 | **93.4** | 89.0 | **93.0** | 61.0 | **77.0** | **92.3** | **63.0** | **77.6** |

## B. Additional Experimental Results

We further extend the experimental evaluation presented in the main paper to demonstrate the robustness and superiority of our cognitive visual search method.

### B.1. Comparison with Additional High-Resolution Image Processing Methods

We extend our evaluation by comparing *CVSearch* with a broader range of recent high-resolution image perception methods reported in Table 7. DC$^2$ (Wang et al., 2025c) is a train-free method that constructs an image tree and synthesizes textual descriptions from leaf patches up to the root to supplement visual information. VisCrop (Zhang et al., 2025a) is also a training-free method that re-feeds the region focused by the attention map to enable a "look again" mechanism. Pixel Reasoner (Wang et al., 2025b) and DeepEyes (Zheng et al., 2025) use curated reasoning trajectories and curiosity-driven reinforcement learning to learn zooming policies for fine-grained reasoning.

Compared to DC$^2$ and VisCrop, our method achieves a substantial advantage, which suggests that supplementing visual features with text summarization (DC$^2$) or relying on a single attention-based crop (VisCrop) is insufficient for capturing the intricate details required for fine-grained perception. Furthermore, compared to RL-based methods like DeepEyes and Pixel Reasoner, our training-free approach achieves superior accuracy. LLaVA-OV-7B with *CVSearch* reaches a *V* Bench* accuracy of 91.6, surpassing DeepEyes without requiring curated data or extensive training. This trend persists across *HR-Bench 4K* and *HR-Bench 8K*. By empowering general MLLMs solely through a dedicated inference-time cognitive search mechanism, *CVSearch* demonstrates that high-performance visual search can be achieved without the complexity and cost of reinforcement learning pipelines.

### B.2. Ablation Study on Information Sufficiency Threshold $\tau_q$

We analyze the sensitivity of our cognitive *Assess-then-Search* mechanism to the threshold $\tau_q$ in Figure 5. In Figure 5a, raising $\tau_q$ serves as a stricter gatekeeper for the *Direct Answer* mode. We observe a monotonic decrease in direct responses and a corresponding increase in *Expert* and *Scan* searches. The results confirm that the model correctly interprets higher $\tau_q$ as a demand for more visual evidence, effectively operationalizing **cognitive gating**. In Figure 5b, this cautious behavior translates into overall performance gains. Notably, the *Direct Answer* accuracy exhibits a slight non-monotonic trend as $\tau_q$ rises. This fluctuation is a natural artifact of shifting sample distributions rather than poor calibration: as the threshold increases, many "easy" queries are conservatively routed to search modules, temporarily concentrating overconfident errors in the shrinking direct answer pool. Crucially, the *Direct Answer* accuracy consistently remains high (above 91.6% across all thresholds), demonstrating that $\tau_q$ reliably isolates simpler queries. Consequently, while the accuracy of *Scan-Search* remains relatively stable, the *Overall* system accuracy improves as $\tau_q$ approaches 0.9. This confirms that suppressing uncertain direct answers and redirecting them to finer-grained search modes is crucial for maximizing performance on challenging benchmarks.

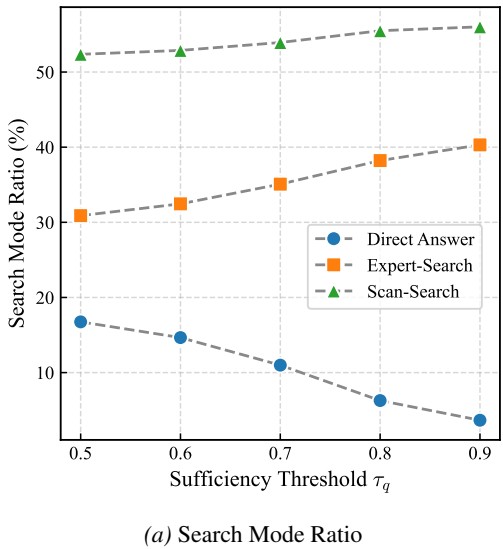

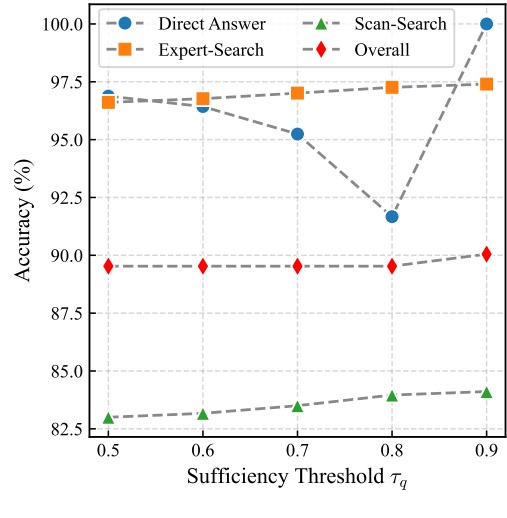

*(a)* Search Mode Ratio

*(b)* Accuracy Comparison

*Figure 5.* Ablation study on the information sufficiency threshold $\tau_q$ on *V\* Bench*. Evaluated with Qwen2.5-VL-7B, we analyze (a) the usage ratio of different search modes and (b) their corresponding accuracy as $\tau_q$ varies from 0.5 to 0.9.

## C. Qualitative Analysis and Case Studies

To provide intuitive insights into the operational mechanisms of **CVSearch**, this section presents qualitative visualizations on challenging samples. We first demonstrate how our *Semantic Guided Adaptive Patching* strategy adaptively partitions images based on visual complexity. Subsequently, we provide a detailed analysis of representative successful inference examples as well as typical failure cases to discuss the current capabilities and limitations of our approach.

### C.1. Visualization of Semantic Guided Adaptive Patching

In this subsection, we compare the image patching mechanisms of our *Semantic Guided Adaptive Patching (SGAP)* strategy against the methods employed by Zoom Eye (Shen et al., 2025) and RAP (Wang et al., 2025d). Figure 6, 7, 8, 9 visualize how these distinct paradigms partition high-resolution images across varying scenarios.

**Rigid Partitioning and Fragmentation.** Existing methods typically rely on semantic-agnostic rules. Zoom Eye utilizes a hierarchical rigid partitioning scheme that divides images into fixed grid layouts, while RAP adopts a sliding window approach with a fixed patch size. As observed in the visualizations, these rigid strategies often sever objects at arbitrary positions. For example, the church in Figure 7 is shattered into numerous disjoint tokens by the dense patching of RAP, and the truck in Figure 8 is split centrally by the fixed grid of Zoom Eye. This disruption of spatial coherence forces the MLLM to reconstruct object semantics from fragmented pieces.

**Semantic-Preserving Adaptivity.** In contrast, **CVSearch** leverages visual features to dynamically determine patch boundaries. Our *SGAP* mechanism groups regions based on semantic similarity to ensure patches naturally align with object contours rather than cutting through them. As shown in the visualizations, we present the first-layer results and highlight specific regions (red dashed boxes) to demonstrate the second-layer refinement. This approach effectively preserves the semantic integrity of targets while reducing token redundancy. Crucially, *SGAP* also provides a quantifiable **Visual Complexity Score** (annotated above each patch in the figures), which serves as an indicator to distinguish information-dense regions from redundant backgrounds. This metric drives the hierarchical process, guiding the model to intensively explore high-value areas while pruning low-complexity regions like the sky, effectively preserving semantic integrity while reducing token redundancy.

### C.2. Success and Failure Case Analysis

In this subsection, we conduct a qualitative analysis of **CVSearch** using the Qwen2.5-VL-7B backbone on the **V\* Bench**. We categorize the inference process into different scenarios to demonstrate how our *Assess-then-Search* workflow adaptively allocates computational resources based on visual difficulty.

**Adaptive Efficiency ( Figure 10)**. *CVSearch* effectively distinguishes task difficulty. For visually prominent targets (*e.g.,* the cap), the *Global Assessment* module enables a Direct Answer, bypassing unnecessary searching. For small but perceptible objects in cluttered scenes (*e.g.,* the stool), it triggers Visual Expert Assisted Search, utilizing SAM 3 for rapid localization without the overhead of full scanning.

**Robustness via Iterative Search (Figure 11).** For extremely tiny targets where initial attempts fail, *CVSearch* employs an iterative cycle by feeding the optimal candidate region back into the Visual Expert. Figure 11 illustrates illustrates this process. In the "tissue box" case (Left), the Expert successfully localizes the target within the zoomed candidate. In the more challenging "helmet" case (Right), the Expert fails again even with the zoomed view, triggering the system to activate Scene-aware Scanning to meticulously capture the visual evidence via bottom-up exploration.

**Failure Analysis (Figure 12).** We examine errors where localization succeeds but reasoning fails. **Left (Hallucination):** The Expert precisely locates the "SUV", yet the MLLM misidentifies its color as "blue" (GT: Silver), revealing a misalignment between visual evidence and internal representation. **Right (Ambiguity):** Iterative search captures the "clock", but the model describes the face ("white") while the ground truth refers to the frame ("green"), highlighting challenges with multi-part object attributes.

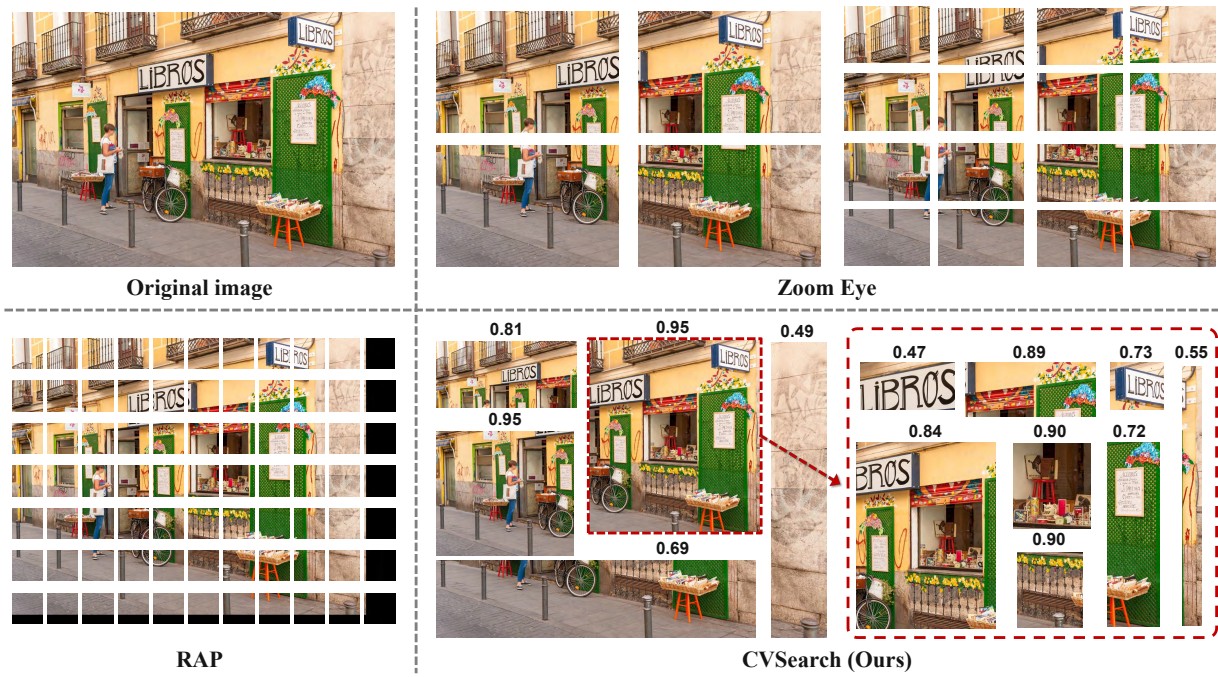

*Figure 6.* **Comparison of patching strategies on a text-rich scene.** Zoom Eye and RAP impose rigid grids that sever the storefront sign ("LIBROS") and the entrance, disrupting OCR and scene understanding. In contrast, our *CVSearch* adaptively partitions the image based on semantic coherence. The annotated values represent **Visual Complexity Scores**. The high visual complexity score (0.95) of the central storefront triggers a focused, boundary-preserving partition in the second layer (red dashed box), keeping the text and objects intact.

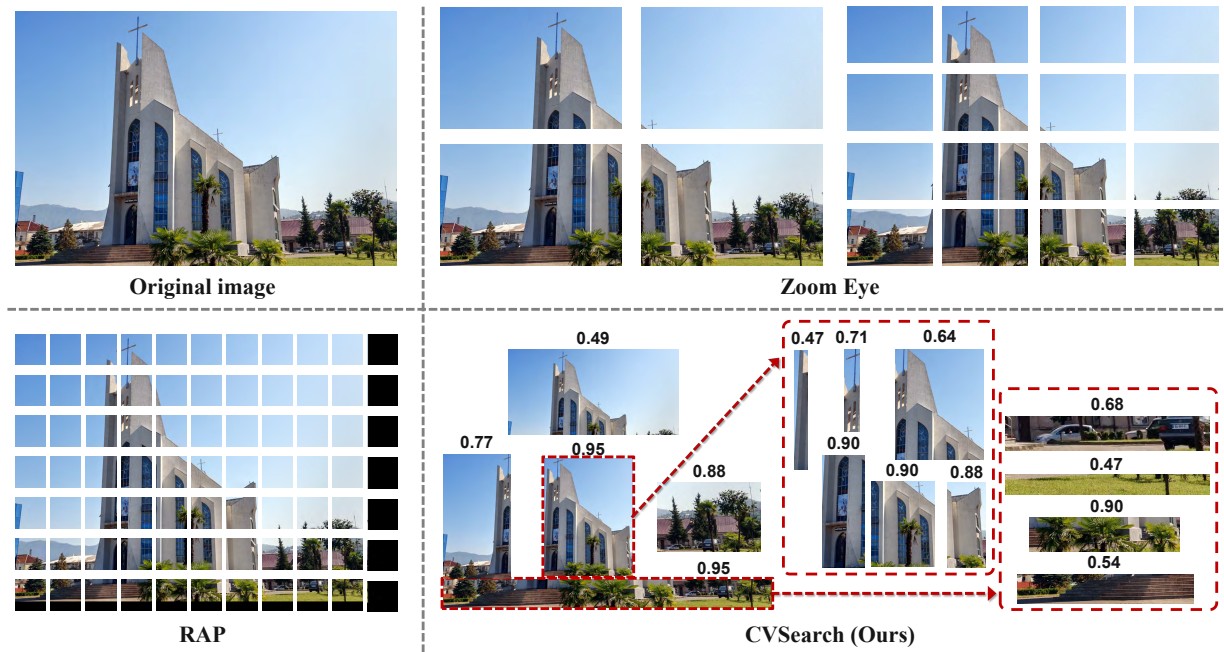

*Figure 7.* **Visualization of semantic preservation in architectural scenes.** Rigid partitioning methods (Zoom Eye and RAP) fragment the continuous structure of the church into disjoint blocks, separating the spire from the nave. *CVSearch* effectively separates the foreground architecture from the low-complexity sky background (0.49). The annotated values represent **Visual Complexity Scores**. The adaptive patching respects the building's geometry, ensuring the main structure is encapsulated within semantically consistent regions.

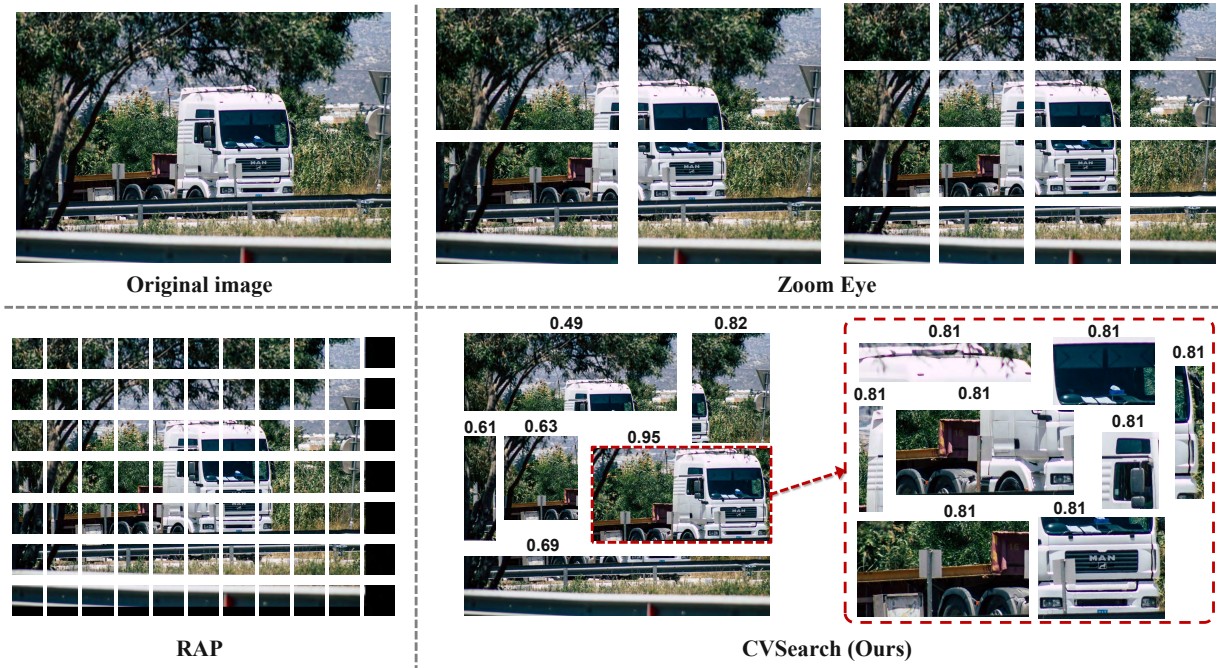

*Figure 8.* **Impact of patching on object integrity.** In the Zoom Eye and RAP examples, the truck is arbitrarily sliced by grid lines, making it difficult to perceive the vehicle as a whole. *CVSearch* utilizes semantic clustering to maintain the integrity of the truck cabin and the surrounding environment. The annotated values represent **Visual Complexity Scores**. The resulting patches group the vehicle features together while separating them from the background trees, facilitating more accurate object recognition.

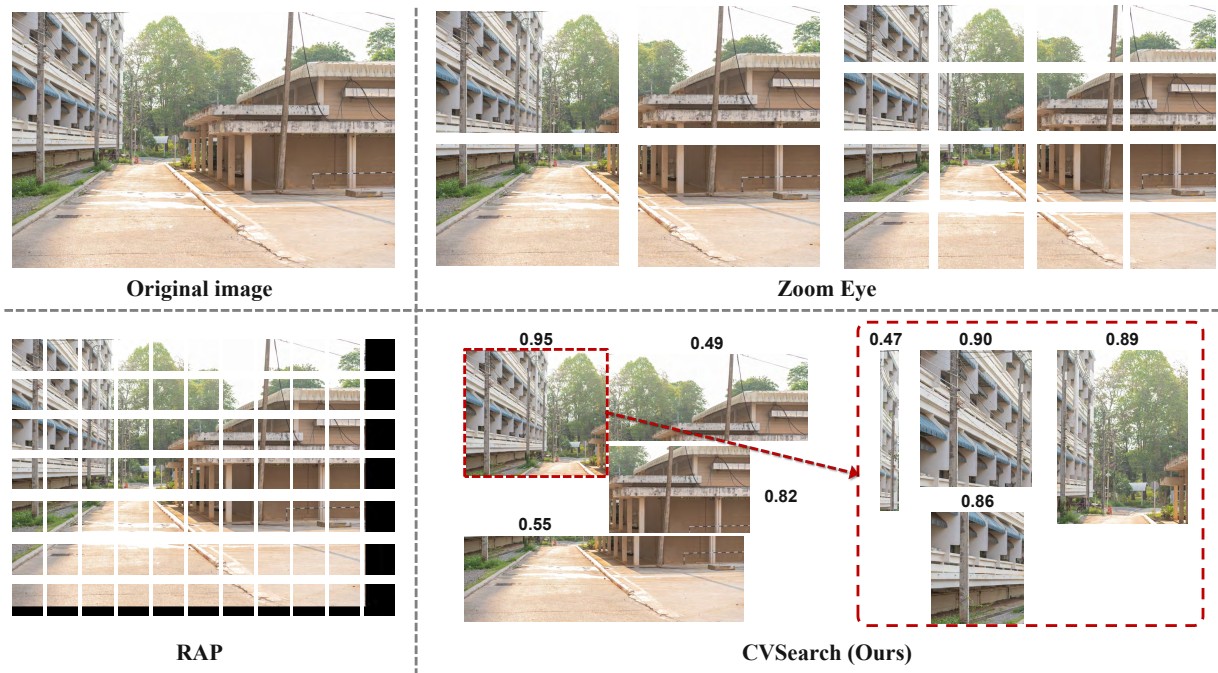

*Figure 9.* **Comparison in cluttered scenarios.** While rigid grids (Zoom Eye, RAP) indiscriminately divide the scene, *CVSearch* demonstrates superior flexibility. The annotated values represent **Visual Complexity Scores**. By calculating visual complexity scores to identify information-dense regions, our method ensures detailed scrutiny where necessary while pruning low-complexity background areas to maintain efficiency.

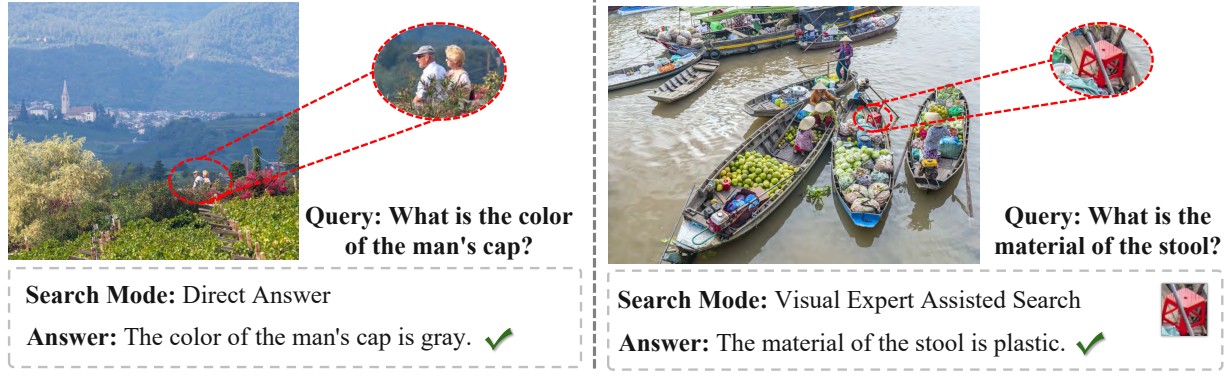

*Figure 10.* **Adaptive search modes for efficiency. Left:** For prominent targets, *CVSearch* employs **Direct Answer** to minimize latency. **Right:** For small objects, it activates **Visual Expert Assisted Search** for precise localization, avoiding the cost of exhaustive scanning.

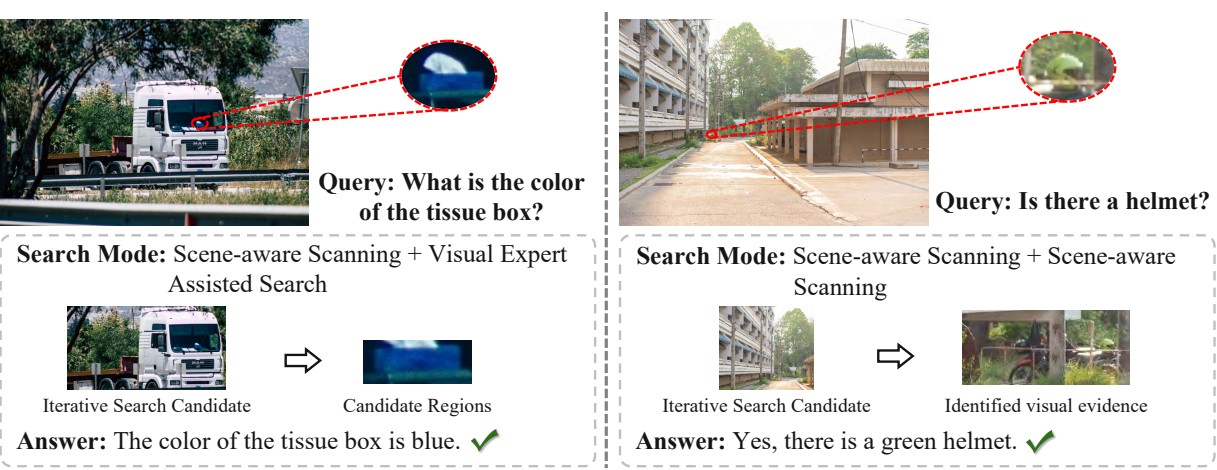

*Figure 11.* **Iterative Search for hard samples.** When initial searches fail, the system zooms into the best candidate. **Left:** The enhanced resolution enables the **Visual Expert** to detect the "tissue box". **Right:** For the extremely small "helmet", the Expert fails again, but the fine-grained **Scene-aware Scanning** successfully captures the target in the second round.

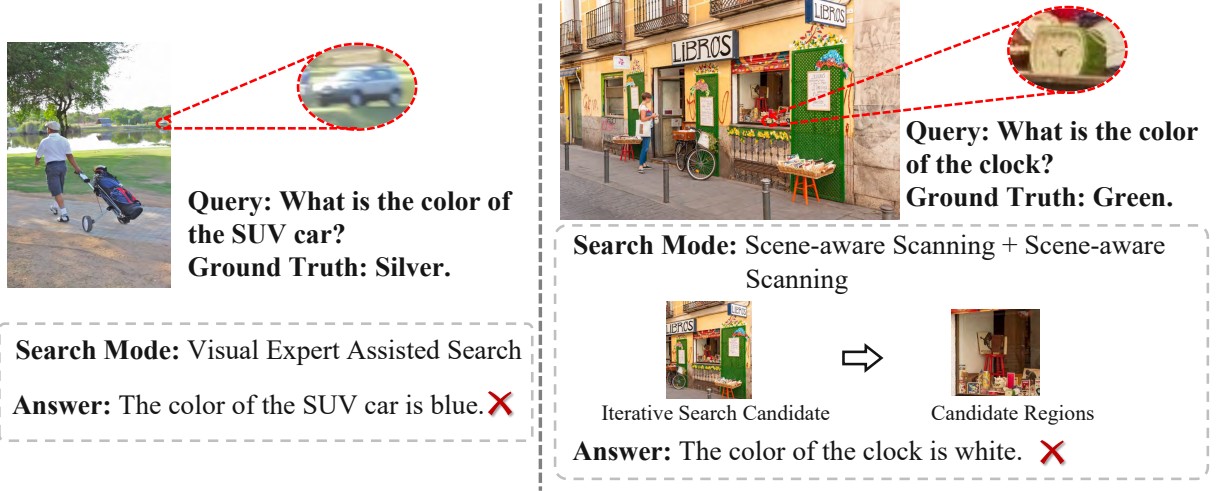

*Figure 12.* **Failures despite accurate localization. Left:** MLLM hallucinates the car color despite correct expert cropping. **Right:** Answer diverges due to attribute ambiguity (describing the clock face instead of the frame).

