# OpenReview forum: "CVSearch: Empowering Multimodal LLMs with Cognitive Visual Search for High-Resolution Image Perception"
_ICML.cc/2026/Conference — ICML 2026 regular_

### Official Review · Reviewer_pcro · 2026-02-20

**Soundness:** 2
**Presentation:** 3
**Significance:** 3
**Originality:** 2
**Overall Recommendation:** 4
**Confidence:** 4

**Summary:**

This paper proposes CVSearch, a training-free visual search framework designed to improve high-resolution (HR) perception in multimodal large language models (MLLMs). The method introduces an Assess-then-Search workflow that dynamically switches between direct answering, visual expert–assisted search (via SAM-based proposals), and scene-aware scanning with semantic-guided adaptive patching and bottom-up exploration. The framework aims to balance efficiency and robustness in HR perception without retraining backbone models. Experiments across multiple HR benchmarks demonstrate substantial accuracy gains and improved inference efficiency compared to prior expert-assisted and scan-based approaches.

**Compliance With Llm Reviewing Policy:**

Affirmed.

**Final Justification:**

This paper presents a strong and practical system for high-resolution perception in MLLMs, with consistent empirical improvements across multiple benchmarks and models. The training-free design and efficiency analysis further enhance its practical relevance.

The rebuttal effectively addresses most of my concerns. In particular, the additional analyses for the confidence-based gating mechanism and the relationship between search modes and sample difficulty improve the credibility of the approach. I also appreciate the authors’ clarification regarding the “adaptive configuration,” which will be revised for better clarity.

Overall, while the contribution remains primarily system-level with moderate algorithmic novelty, the method is well-engineered and practically useful. I therefore maintain my weak accept recommendation.

**Key Questions For Authors:**

- **Throughput behavior in Table 6.**
    - In Table 6, introducing the global assessment (e.g., with SAM 3 or additional search modules) appears to *reduce* throughput compared to the vanilla baseline. Intuitively, if global assessment allows bypassing expensive scanning, one might expect faster inference. Could the authors clarify why throughput decreases at this stage? Is the global assessment itself computationally dominant, or is there additional overhead not discussed? A breakdown of latency per module would help.

- **“Adaptive configuration” vs. hyperparameter selection (Tree depth).**
    - The paper describes the tree depth setting (D = 2 for single-object, D = 3 for multi-object) as an adaptive configuration (Line 431). However, this appears to be a manually defined rule based on target count rather than a dynamically learned or data-driven adaptation. Could the authors clarify in what sense this is adaptive, and whether depth selection is sensitive to dataset or query type?

- **Relationship between search mode accuracy and sample difficulty (Figure 4 vs. Appendix Figure 5b).**
    - The paper suggests that different search modes align with different levels of sample difficulty. However, since the threshold τq directly controls how many samples are routed to each mode, changing τq alters the distribution of problems each mode handles. Therefore, mode-wise accuracy alone may not directly reflect intrinsic difficulty. Could the authors clarify whether they controlled for this confound, and whether there is evidence that samples handled by Scan-Search are genuinely harder rather than simply threshold-routed?

**Limitations:**

The paper does not include a dedicated limitations section in the main text, although failure cases are provided in the appendix.

**Strengths And Weaknesses:**

- Strengths
    - Addresses an important and well-motivated problem: the perceptual bottleneck of MLLMs under high-resolution inputs.
    - Presents a **strong practical system** that is training-free and consistently improves multiple backbone models.
    - Demonstrates large empirical gains across diverse benchmarks, with efficiency comparisons included.
- Weaknesses
    - **The core decision mechanisms are insufficiently validated.:** Both the confidence-based information sufficiency gating and the expert proposal verification are central to the framework’s claimed efficiency–robustness balance, yet neither is rigorously analyzed.
        - *Confidence-based gating.* While Appendix B.2 provides threshold sensitivity results for τq, the paper does not directly demonstrate that the confidence score (Eq. 2) reliably predicts answer correctness. There is no correlation analysis, calibration study, or breakdown of false direct accept vs. false reject cases. As a result, it remains unclear whether the gating signal is genuinely predictive or primarily tuned via threshold selection.
        - *Expert proposal verification.* The criteria used to determine whether SAM-generated bounding boxes “adequately cover” target objects are not clearly formalized. The paper does not quantify proposal quality (e.g., coverage accuracy, false acceptance/rejection rates) or analyze how incorrect acceptance impacts downstream robustness. Since incorrect verification may block fallback scanning, this module critically affects system reliability.
    - **Algorithmic novelty is moderate.:** Although the system is practically effective, many components (superpixel segmentation, agglomerative clustering, tree search, confidence routing) are established techniques. The cognitive framing is intuitive but not theoretically grounded, and the contribution primarily lies in system integration rather than new learning principles.

---

> ### Author Rebuttal · Authors · 2026-03-31
>
> We thank you for recognizing our work's **strong practical significance** and **large empirical gains**. We address your concerns point by point.
>
> ### pcro-W1
> > The core decision mechanisms are insufficiently validated.
>
> **1. Confidence-based Gating**:
>
> **Correlation Analysis**: We evaluated the statistical correlation between $c_q$ scores and vanilla Qwen2.5-VL-7B correctness on V* dataset, yielding an **AUC-ROC score of 0.7760** and a **Point Biserial Correlation (PBC) of 0.3344 ($p = 2.27 \times 10^{-6}$)**. This confirms $c_q$ reliably predicts correctness, rather than being a mere tuned hyperparameter.
>
> **Breakdown of Gating Decisions**: We analyze the routing decisions at the threshold $\tau_q = 0.90$ (Qwen2.5-VL-7B on V*). **Table A** shows a 0.0% False Accept rate, ensuring safety against overconfident errors. The 69.1% False Reject rate is an intentional trade-off: routing low-confidence queries to search adds compute but avoids unrecoverable errors.
>
> **Table A: Gating Decisions ($\tau_q = 0.90$)**
> | System Decision | Correct Ratio | Incorrect Ratio |
> | :--- | :---: | :---: |
> | Direct Answer ($c_q \ge \tau_q$) | 4.2% (True Accept) | 0.0% (False Accept) |
> | Trigger Search ($c_q < \tau_q$) | 69.1% (False Reject) | 26.7% (True Reject) |
>
> **2. Expert Proposal Verification**
>
> **Formalizing "Adequate Coverage"**: We clarify this is met when the number of targets (across different categories) segmented by SAM 3 strictly matches the number of extracted target objects.
>
> **Proposal Quality and Downstream Impact**: To analyze the impact of imprecise proposals, we evaluated Qwen2.5-VL-7B w/ CVSearch on V* using ground-truth bounding boxes. We analyzed the samples routed to the Visual Expert Assisted mode based on the IoU of SAM's generated proposals. **Table B** confirms robustness: even for low-quality proposals (IoU < 0.5), accuracy remains 92.9% (vs. 98.4% for IoU $\ge$ 0.5). This resilience stems from our deliberate design of enlarging SAM bounding boxes during cropping to ensure sufficient context.
>
> **Table B: Proposal Quality & Acc. (Expert Assisted)**
> | Proposal Quality | Sample Ratio (%) | Accuracy (%) |
> | :--- | :---: | :---: |
> | IoU ≥ 0.5 | 81.6 | 98.4 |
> | IoU < 0.5 | 18.4 | 92.9 |
> | Overall | 100.0 | 97.4 |
>
> ### pcro-W2
> > Algorithmic novelty is moderate.
>
> Thank you for recognizing the practical effectiveness and intuitive cognitive framing of our framework. While individual components are established, our core contribution is a system-level innovation. We orchestrate these techniques into a cohesive, adaptive pipeline that fundamentally enhances MLLM visual reasoning without the massive costs of retraining or reinforcement learning. The novelty lies in this architectural integration: how confidence routing and tree search dynamically bridge high-resolution visual inputs and MLLM cognition.
>
> ### pcro-Q1
> > Throughput behavior in Table 6
>
> **Clarification of Vanilla Baseline**: In Table 6, the vanilla baseline only uses Direct Answer mechanism; introducing Global Assessment naturally adds latency relative to doing nothing extra.
>
> **Per-Module Latency**: We break down the average latency for Qwen2.5-VL-7B w/ CVSearch on V* dataset. **Table C** shows Global Assessment (1.37s) is not the bottleneck. The primary overhead is the Bottom-up Search (14.68s), which is triggered only for highly complex samples. Notably, the different input image resolutions cause the Answer Generation time discrepancy between the two models.
>
> **Table C: Latency Breakdown for CVSearch (s)**
> | Model | Global Assess. | Visual Cue Gen. | Visual Expert | Tree Const. | Bottom-up Search | Answer Gen. |
> | :--- | :---: | :---: | :---: | :---: | :---: | :---: |
> | Vanilla Backbone | 0 | 0 | 0 | 0 | 0 | 5.56 |
> | w/ CVSearch | 1.37 | 1.75 | 0.25 | 0.55 | 14.68 | 1.39 |
>
> ### pcro-Q2
> > “Adaptive configuration” vs. hyperparameter selection (Tree depth).
>
> We clarify "adaptive" refers to **sample-level dynamic routing**, not a learned or fixed global hyperparameter. CVSearch dynamically adapts to per query: D=2 for single target (saving compute) and D=3 for multiple targets (capturing context).
>
> ### pcro-Q3
> > Relation between search mode accuracy and sample difficulty
>
> To prove Scan-Search tackles genuinely harder queries, rather than just threshold-routed ones, we evaluated Qwen2.5-VL-7B w/ CVSearch on V* dataset, using **relative target size ($\sqrt{W \times H}$) as an objective difficulty metric (Smaller is harder)**. **Table D** shows assigned modes align with actual difficulty, dictating accuracy gains. This confirms our gating dynamically targets genuine visual complexity, not mere threshold selection.
>
> **Table D: Sample Difficulty & Performance Gain**
> | Search Mode | Avg Target Size | Backbone Acc (%) | w/ CVSearch Acc (%) | Gains (%) |
> | :--- | :---: | :---: | :---: | :---: |
> | Direct Answer | 80.2 | 100.0 | 100.0 | +0.0 |
> | Visual Expert Assisted | 66.3 | 92.1 | 97.4 | +5.3 |
> | Scene-aware Scanning | 48.7 | 57.9 | 81.3 | +23.4 |

---

> > ### Author Rebuttal · Reviewer_pcro · 2026-04-03
> >
> > The rebuttal addresses most of my concerns and provides useful additional evidence, particularly for the gating mechanism and search mode analysis, which improves the credibility of the approach.
> > One minor point is that the “adaptive configuration” for tree depth appears to be more of a rule-based design choice, and the wording could be clarified.
> > Overall, I appreciate the authors’ clarifications, and my score remains unchanged.

---

> > > ### Author Response · Authors · 2026-04-03
> > >
> > > Thank you for reviewing our rebuttal and for your continued support of our work. We are very pleased that our added analyses successfully resolved your concerns.
> > >
> > > Regarding your minor point on the "adaptive configuration," we sincerely appreciate the correction. We will modify the terminology in the final manuscript to explicitly state that this is a rule-based design choice, ensuring perfect clarity. All other new evidence and discussions will also be carefully incorporated.

---

### Official Review · Reviewer_Z2i8 · 2026-03-06

**Soundness:** 3
**Presentation:** 3
**Significance:** 2
**Originality:** 3
**Overall Recommendation:** 4
**Confidence:** 4

**Summary:**

The paper proposes CVSearch, a training-free adaptive visual search framework aimed at addressing the tension faced by multimodal large language models (MLLMs) when perceiving high-resolution (HR) images, namely the trade-off between full coverage and efficiency. The method introduces a cognitive-psychology-inspired “evaluate before search” workflow, adaptively switching between expert-assisted localization and semantic-aware scanning, and substantially improving the detection of tiny targets.

**Compliance With Llm Reviewing Policy:**

Affirmed.

**Final Justification:**

My concerns are resolved and I will retain my positive scores accordingly

**Key Questions For Authors:**

**Iteration depth:** The experiments suggest limited gains but increased cost at depth (D=4). Is this conclusion resolution-dependent, and does it still hold for 16K or higher-resolution imagery?

**Expert generality:** The experiments mainly use SAM 3. If replaced with a lighter or architecturally different visual expert, do the gains remain robust?

**Failure modes:** The paper mentions reasoning failures. Could the authors summarize which task types (e.g., long-tail object recognition or detecting extremely thin structures) remain challenging for the current framework?

**Limitations:**

yes

**Strengths And Weaknesses:**

**Strengths.**

1.The paper builds a logically coherent computational model grounded in dual-pathway theories of human vision, combining non-selective global perception with selective local inspection.  The proposed SGAP mitigates the semantic fragmentation caused by rigid grid-based scanning by clustering visual-expert features to preserve object-level semantic integrity.

3.As a plug-and-play approach, CVSearch generalizes well across multiple MLLM backbones (e.g., Qwen2.5-VL and InternVL2.5), achieving state-of-the-art accuracy on several HR benchmarks such as V*Bench and HR-Bench.

**Weaknesses:**

1.computational cost: While the paper emphasizes efficiency gains from branch pruning, it remains unclear whether computational cost degrades to that of conventional scanning methods under highly complex scenes (e.g., dense crowds or textured backgrounds). A latency analysis under extreme visual complexity would strengthen the claims.

 2.Dependence on the visual expert: SGAP relies on deep features from a visual expert (e.g., SAM 3). If the expert produces low-quality features under challenging conditions such as extreme lighting or low contrast, it may mislead downstream semantic clustering.

---

> ### Author Rebuttal · Authors · 2026-03-30
>
> We are encouraged by your recognition of our **logically coherent computational model**, **the effectiveness of SGAP**, and our work's **strong practical significance**. We address your concerns below.
>
> ### Z2i8-W1
> > computational cost:  computational cost in highly complex scenes
>
> CVSearch maintains high efficiency even in highly complex scenes. HR-8K benchmark serves as a stress test for such conditions, featuring massive visual tokens, dense object clusters, and intricate backgrounds. Table 3 shows CVSearch avoids conventional scanning costs on HR-8K, achieving a 1.92 throughput, a 2.8x-3.3x speedup over Zoom Eye (0.68) and RAP (0.58). Three designs ensure this efficiency:
> * **Visual Expert-Led Efficiency**: CVSearch prioritizes SAM 3. Its superior object localization often resolves queries directly, completely bypassing the costly iterative scanning.
> * **Semantic Integrity via SGAP**: Unlike grid patching used in conventional methods that fragments objects and causes repetitive search, SGAP preserves object-level integrity, drastically reducing required search steps.
> * **Prior-Guided Search Priority**: SGAP provides a visual complexity prior. Instead of evaluating every patch with the heavy MLLM, CVSearch strictly prioritizes high-value regions.
>
> Thus, CVSearch empirically and theoretically avoids degrading to exhaustive scanning costs.
>
> ### Z2i8-W2
> > Dependence on visual expert: SAM 3 features under challenging lighting conditions
>
> While extreme conditions may degrade SAM 3 features, CVSearch prevents catastrophic failures via systemic robustness:
> * **Graceful Degradation**: If noisy features compromise SGAP's clustering, it merely degrades to a state comparable to conventional grid-based patching (which blindly cuts objects).
> * **MLLM as the Ultimate Anchor**: SGAP only provides search priors. The robust MLLM performs the final reasoning, effectively contextualizing fragmented patches. Thus, clustering errors primarily reduce search efficiency (more steps) rather than severely dropping final accuracy.
>
> ### Z2i8-Q1
> > Iteration depth: the optimal tree depth
>
> The optimal tree depth is resolution-dependent. The limited gains at D=4 are specific to $\le$ 8K resolutions. For 16K or higher-resolution imagery, the massive search space makes D=3 too coarse, rendering deeper depth (D$\ge$4) essential for microscopic details. Crucially, deeper bottom-up search does not explode costs. Our SGAP and Branch Pruning aggressively filter irrelevant background nodes before MLLM inference, concentrating compute exclusively on high-value regions to balance ultra-fine extraction with practical efficiency.
>
> ### Z2i8-Q2
> > Expert generality: different visual expert
>
> To evaluate generality, we replaced SAM 3 with GroundingDINO, a lighter detection-based model.
>
> **Robustness Across Backbones**: **Table A** shows replacing SAM 3 with GroundingDINO yields consistent gains across various backbones, proving our framework isn't tied to a specific expert.
>
> **Table A: Impact of Different Visual Experts**
> | Backbone | Search Method | V* | HR-4K | HR-8K |
> | :--- | :--- | :--- | :--- | :--- |
> | Qwen2.5-VL-7B | Direct Answer | 71.2 | 68.8 | 65.3 |
> | | w/ CVSearch (GroundingDINO) | 88.0 | 72.8 | 71.1 |
> | | w/ CVSearch (SAM 3) | 90.1 | 76.6 | 75.6 |
> | Qwen3-VL-2B | Direct Answer | 79.1 | 71.3 | 67.8 |
> | | w/ CVSearch (GroundingDINO) | 90.1 | 73.9 | 70.4 |
> | | w/ CVSearch (SAM 3) | 92.2 | 74.0 | 73.6 |
> | Qwen3-VL-4B | Direct Answer | 89.5 | 76.3 | 71.6 |
> | | w/ CVSearch (GroundingDINO) | 93.2 | 77.5 | 76.3 |
> | | w/ CVSearch (SAM 3) | 93.7 | 77.4 | 75.1 |
> | Qwen3-VL-8B | Direct Answer | 88.0 | 79.1 | 74.5 |
> | | w/ CVSearch (GroundingDINO) | 90.6 | 79.5 | 78.4 |
> | | w/ CVSearch (SAM 3) | 91.6 | 79.5 | 76.5 |
>
> **Dynamic Compensatory Mechanism**: Table B reveals why performance stays robust. With the weaker GroundingDINO, queries resolved by the "Visual Expert Assisted" module drop (40.3% $\rightarrow$ 24.6%). However, CVSearch adaptively routes unresolved queries to "Scene-aware Scanning", increasing its usage (56.0% $\rightarrow$ 71.7%). This reliable fallback absorbs the lighter expert's shortcomings, maintaining competitive overall accuracy.
>
> **Table B: Search Dynamics of Different Visual Experts (Qwen2.5-VL-7B on V\*)**
> | Visual Expert | Metric | Direct Answer | Visual Expert Assisted | Scene-aware Scanning |
> | :--- | :--- | :--- | :--- | :--- |
> | GroundingDINO | Search Ratio | 3.7 | 24.6 | 71.7 |
> | | Accuracy | 100.0 | 91.3 | 86.1 |
> | SAM 3 | Search Ratio | 3.7 | 40.3 | 56.0 |
> | | Accuracy | 100.0 | 97.4 | 84.1 |
>
> ### Z2i8-Q3
> > Failure modes: task types remain challenging
>
> Our error analysis identifies three primary challenging task types: (1) Text/Semantic Logic: Tasks requiring intensive OCR or complex mathematical reasoning. (2) Knowledge Gaps: Tasks requiring the recognition of specialized "long-tail objects" beyond the MLLM's parametric knowledge. (3) Perception Limits: Tasks requiring the localization of extremely small targets.

---

> > ### Author Rebuttal · Reviewer_Z2i8 · 2026-04-01
> >
> > My concerns are resolved and I will continue my positive scores accordingly.

---

> > > ### Author Response · Authors · 2026-04-03
> > >
> > > Thank you for your positive response and for maintaining your positive scores. We greatly appreciate your constructive suggestions and will make sure the new results and discussions from the rebuttal are integrated into the final manuscript.

---

### Official Review · Reviewer_S5R1 · 2026-03-07

**Soundness:** 3
**Presentation:** 3
**Significance:** 3
**Originality:** 3
**Overall Recommendation:** 4
**Confidence:** 4

**Summary:**

1.This paper proposes CVSearch, a training-free framework for high-resolution image perception in MLLMs.

2.The main idea is an Assess-then-Search pipeline: answer directly when global information is sufficient, use expert-assisted search when needed, and switch to semantic-aware scanning if expert proposals fail.

3.The method introduces Semantic Guided Adaptive Patching (SGAP) and Dynamic Bottom-Up Search to preserve object semantics and reduce redundant scanning.

4.Experiments show consistent gains across multiple backbones and benchmarks, with strong efficiency as well.

**Compliance With Llm Reviewing Policy:**

Affirmed.

**Final Justification:**

The author addressed my main concerns and I tend to accept this paper.

**Key Questions For Authors:**

1. In Fig.5(b), why does the Direct Answer accuracy first decrease and then increase as c_q rises? This non-monotonic trend suggests that the sufficiency score c_q may not be well calibrated as a routing signal.

**Limitations:**

Lack of limitations and negative societal impact.

**Strengths And Weaknesses:**

Strengths:

1.The framework is well-motivated and coherent, with a clear failure-aware design from global assessment to expert search to semantic scanning.

2.Strong empirical results: e.g., LLaVA-OV-7B improves from 75.4 to 91.6 on V* Bench, and InternVL2.5-8B reaches 77.6 on HR-Bench 8K with CVSearch.

3.Good efficiency: CVSearch achieves the best reported throughput among compared search methods, including over 3× speedup over RAP on HR-8K.


Weaknesses:

1.The paper does not compare against larger and stronger MLLMs, which limits the claim of scalability. Although the authors include models up to Qwen3-VL-8B and show consistent gains across 2B–8B variants, it remains unclear whether CVSearch would still provide substantial improvements on much stronger models such as Qwen3-VL-32B, Qwen3-VL-30B-A3B, or even Qwen3-VL-235B-A22B. More importantly, as the native perception ability of the base model improves, the incremental benefit of an additional search framework may shrink.

2.The efficiency evaluation is still somewhat incomplete. The paper reports throughput comparisons only against other search-based methods, however, the paper does not clearly quantify the average inference-time overhead of enabling CVSearch relative to the vanilla backbone itself on the same benchmarks. Since CVSearch adds gating, expert search, and iterative scanning, it would be useful to know how much slower w/ CVSearch is than the original model in practice.

---

> ### Author Rebuttal · Authors · 2026-03-30
>
> We sincerely thank you for the constructive feedback. We are encouraged by your positive assessment of our work, particularly your recognition of **the well-motivated and coherent design of our failure-aware framework**, our **strong empirical results** , and **the superior efficiency**. We also appreciate your valuable feedback regarding the scalability of our framework and the computational overhead relative to the vanilla backbones. In the following section, we address your specific concerns point by point.
>
> ### S5R1-W1
> > The paper does not compare against larger and stronger MLLMs, which limits the claim of scalability. Although the authors include models up to Qwen3-VL-8B and show consistent gains across 2B–8B variants, it remains unclear whether CVSearch would still provide substantial improvements on much stronger models such as Qwen3-VL-32B, Qwen3-VL-30B-A3B, or even Qwen3-VL-235B-A22B. More importantly, as the native perception ability of the base model improves, the incremental benefit of an additional search framework may shrink.
>
> We appreciate this insightful point. While exhaustively evaluating massive models (e.g., 235B) exceeds our compute limits (an 8× A100 node) during the short rebuttal period, we have successfully scaled our framework to the much stronger Qwen3-VL-32B.
>
> As shown in **Table A**, CVSearch continues to improve advanced backbones, boosting Qwen3-VL-32B by +7.5% on HR-8K, +3.8% on HR-4K, and +2.6% on V*. This directly alleviates concerns of diminishing returns, demonstrating that our search mechanism effectively resolves high-resolution bottlenecks even for highly advanced MLLMs.
>
> **Table A: Scaling Performance on Qwen3-VL-32B**
> | Backbone | Search Mode | V* | HR-4K | HR-8K |
> | :--- | :--- | :---: | :---: | :---: |
> | Qwen3-VL-32B | Direct Answer | 86.9 | 76.5 | 70.9 |
> || w/ CVSearch | **89.5** | **80.3** | **78.4** |
>
> ### S5R1-W2
> > The efficiency evaluation is still somewhat incomplete. The paper reports throughput comparisons only against other search-based methods, however, the paper does not clearly quantify the average inference-time overhead of enabling CVSearch relative to the vanilla backbone itself on the same benchmarks. Since CVSearch adds gating, expert search, and iterative scanning, it would be useful to know how much slower w/ CVSearch is than the original model in practice.
>
> We sincerely apologize for the confusion caused by the suboptimal organization of the throughput comparisons in our original manuscript. The throughput comparisons against **the vanilla backbone (Direct Answer)** and **the Expert Assisted method (SAM 3)** were originally provided in **Table 6**.
>
> To explicitly quantify the computational overhead, we have consolidated all measurements into **Table B**. While CVSearch naturally introduces overhead compared to the vanilla backbone, it is significantly faster than other scan-based methods while achieving higher accuracy. To further mitigate this overhead in practice, two strategies are available: 1) System-Level Acceleration: independent nodes at the same depth enable parallel batched inference and Flash Attention can be used to improve base MLLM inference speed; 2) Model-Level Scaling: CVSearch enables lightweight models to match the performance of much larger backbones, offering an exceptional speed-accuracy trade-off.
>
> **Table B: Consolidated Comparison of Accuracy and Throughput**
> | Model | Visual Search Method | V* (Acc) | V* (Thr) | HR-4K (Acc) | HR-4K (Thr) | HR-8K (Acc) | HR-8K (Thr) |
> | :--- | :--- | :---: | :---: | :---: | :---: | :---: | :---: |
> | Qwen2.5-VL-7B | Direct Answer | 71.2 | 8.30 | 68.8 | 7.62 | 65.3 | 7.62 |
> | | w/ Visual Expert (SAM 3) | 84.3 | 3.60 | 71.9 | 5.59 | 68.1 | 5.30 |
> | | w/ Scan-based (Zoom Eye) | 85.3 | 0.68 | 72.5 | 1.29 | 69.8 | 0.68 |
> | | w/ Scan-based (RAP) | 84.8 | 0.66 | 74.8 | 1.22 | 76.0 | 0.58 |
> | | w/ CVSearch | **90.1** | **1.02** | **76.8** | **3.77** | **75.6** | **1.92** |
>
> We will refine Table 3 in the revised manuscript with this consolidated latency comparison.
>
> ### S5R1-Q1
> > In Fig.5(b), why does the Direct Answer accuracy first decrease and then increase as c_q rises? This non-monotonic trend suggests that the sufficiency score c_q may not be well calibrated as a routing signal.
>
> We thank the reviewer for their keen observation. The non-monotonic trend in Fig. 5(b) is a natural artifact of shifting sample distributions, not poor calibration. As the threshold $\tau_q$ increases, "easy" queries are conservatively routed to search modules. This shrinks the remaining Direct Answer pool, temporarily concentrating overconfident errors and causing a slight accuracy dip. Crucially, the Direct Answer accuracy consistently remains above 91.6% across all thresholds, proving $c_q$ reliably isolates simpler queries as an effective routing signal.
>
> ### S5R1-Limitations
> > Lack of limitations and negative societal impact.
>
> We will add a related discussion in our revision.

---

> > ### Author Rebuttal · Reviewer_S5R1 · 2026-04-03
> >
> > Thank you for the detailed and thoughtful responses. These clarifications address my main concerns. I encourage the authors to incorporate these results and discussions clearly in the final version.

---

> > > ### Author Response · Authors · 2026-04-03
> > >
> > > Thank you for your positive response. We sincerely appreciate your constructive feedback and will ensure all the new results and discussions from the rebuttal are integrated into the final manuscript.

---

### Official Review · Reviewer_T216 · 2026-03-10

**Soundness:** 3
**Presentation:** 4
**Significance:** 3
**Originality:** 3
**Overall Recommendation:** 4
**Confidence:** 4

**Summary:**

- Addresses "visual needle in the haystack" problem
- Proposes a training free framework that balances the traid-off between the efficiency of visual expert-asssisted search and coverage of scan-based search
- The framework aims to solve the problem hierarchically, using a mechanism that adaptivly swithces between expert-assisted and scan-based inspired search.
- To increase framework robustness to expert failure, the authors propose *scene-aware scanning mode*
- To address semantic fragmentation of the traditional scanning algorithms the authors propose *semantic guided adaptive patching*
- The authors demonstrate that CVSearch framework leads to a significantly superior performance on high-resolution and general perpose benchmarks for 3 separate backbone VLMs

**Compliance With Llm Reviewing Policy:**

Affirmed.

**Key Questions For Authors:**

Q1. What is the latency of VLMs w/ CVSeach in comparisson to the scan-based and expert-based frameworks?

**Limitations:**

The limitations are not mentioned in the paper. One that I could see is the sometimes prohibitive for some of the applications computation burdain introduced by this approach.

**Strengths And Weaknesses:**

S1: The paper is very well written and provides high-quality visualizations that help understanding the method better

S2: Proposed framework is meaningfull, implementation of all proposed algorithms is well documented (pseudo-code in the Appendix)

S3: Baselines and comparissons provided in the paper in general make sense and support the claims (there are comparissons to open and closed-course VLMs of different sizes, other visual-search frameworks, HR-specializing models. In addition to that numbers for separate addition of all proposed modification are presented, which motivates the necessity of each of them. Throughput numbers for Qwen 2.5-VL-7B are indicated)

W1: The framework is quite computationally expensive (a VLM with CVSearch can be more than 8 times slower than the original VLM). That can be a hindernis for many applications.

W2: Throughput measurements are only provided for one model (Qwen2.5-VL-7B), no latency comparisson with scan-based or expert-based frameworks. This missing comparisson does not allow to validate the claim that the proposed approach is in fact successfully balancing the advantages of two approaches.

---

> ### Author Rebuttal · Authors · 2026-03-30
>
> We sincerely thank you for the constructive feedback. We are encouraged by your positive assessment of our work, particularly your recognition of **the meaningfulness of our proposed framework**, the thoroughness of **our documented implementation**, and **our comprehensive baseline comparisons**. We also appreciate your feedback regarding the need for broader latency comparisons and the clarity of the paper's presentation. In the following section, we address your specific concerns point by point.
>
> ### T216-W1
> > The framework is quite computationally expensive (a VLM with CVSearch can be more than 8 times slower than the original VLM). That can be a hinder for many applications.
>
> We agree that the computational overhead introduced by CVSearch is a crucial factor for real-world deployments. To mitigate this latency bottleneck, we address it from two complementary perspectives:
>
> **System-Level Acceleration**: The search process in CVSearch is highly optimizable. Specifically, in our Dynamic Bottom-Up Search, nodes located at the same depth are independent and can be searched in parallel via batched inference. Furthermore, by integrating memory-efficient mechanisms like Flash Attention, the base MLLM inference speed is significantly improved. As demonstrated in **Table A**, these optimizations effectively slash the latency overhead with only a marginal performance drop. (Efficiency Optimization: Eff. Opt., Accuracy: Acc, Throughput: Thr).
>
> **Table A: Experimental Results of Efficiency Optimization**
> | Model | Visual Search Method | Eff. Opt. | V* (Acc) | V* (Thr) | HR-4K (Acc) | HR-4K (Thr) | HR-8K (Acc) | HR-8K (Thr) |
> | :--- | :--- | :--- | :---: | :---: | :---: | :---: | :---: | :---: |
> | Qwen2.5-VL-7B | Direct Answer | No | 71.2 | 8.30 | 68.8 | 7.62 | 65.3 | 7.62 |
> | | w/ CVSearch | No | 90.1 | 1.02 | 76.8 | 3.77 | 75.6 | 1.92 |
> | | w/ CVSearch | Yes | 89.5 | 4.24 | 75.5 | 6.40 | 75.0 | 2.78 |
>
> **Model-Level Scaling**: For latency-sensitive applications, CVSearch can be seamlessly integrated with lightweight models (e.g., Qwen3-VL-2B/4B). Empowered by CVSearch, these faster models perform on par with much larger ones (see Table 5), offering a highly favorable speed-accuracy trade-off.
>
> We will explicitly include this efficiency analysis in the revised manuscript.
>
> ### T216-W2
> > Throughput measurements are only provided for one model (Qwen2.5-VL-7B), no latency comparison with scan-based or expert-based frameworks. This missing comparison does not allow to validate the claim that the proposed approach is in fact successfully balancing the advantages of two approaches.
>
> We sincerely apologize for the confusion caused by the suboptimal organization of the latency and throughput comparisons in our original manuscript. These measurements were actually conducted and included in the initial submission, but they were unfortunately scattered: the comparison with scan-based frameworks (Zoom Eye and RAP) was presented in Table 3, while the comparison with the expert-based framework (SAM 3) was located in Table 6.
>
> To directly validate our claim of balancing accuracy and efficiency, we have consolidated these throughput measurements into **Table B**. As the unified data demonstrates:
> 1. **Compared to Scan-based methods (Zoom Eye & RAP)**: CVSearch achieves significantly higher accuracy while being substantially faster (e.g., on HR-4K, CVSearch's throughput is 3.77, nearly 3x faster than Zoom Eye's 1.29 and RAP's 1.22).
> 2. **Compared to Expert-based methods (SAM 3)**: While the SAM 3-assisted framework is faster, it suffers from a noticeable drop in accuracy (e.g., 71.9% vs. our 76.8% on HR-4K). CVSearch bridges this gap by delivering state-of-the-art accuracy with a highly competitive throughput.
>
> We will refine Table 3 in the revised manuscript with this consolidated latency comparison.
>
> **Table B: Consolidated Comparison of Accuracy and Throughput**
> | Model | Visual Search Method | V* (Acc) | V* (Thr) | HR-4K (Acc) | HR-4K (Thr) | HR-8K (Acc) | HR-8K (Thr) |
> | :--- | :--- | :---: | :---: | :---: | :---: | :---: | :---: |
> | Qwen2.5-VL-7B | Direct Answer | 71.2 | 8.30 | 68.8 | 7.62 | 65.3 | 7.62 |
> | | w/ Visual Expert (SAM 3) | 84.3 | 3.60 | 71.9 | 5.59 | 68.1 | 5.30 |
> | | w/ Scan-based (Zoom Eye) | 85.3 | 0.68 | 72.5 | 1.29 | 69.8 | 0.68 |
> | | w/ Scan-based (RAP) | 84.8 | 0.66 | 74.8 | 1.22 | 76.0 | 0.58 |
> | | w/ CVSearch | **90.1** | **1.02** | **76.8** | **3.77** | **75.6** | **1.92** |
>
> ### T216-Q1
> > What is the latency of VLMs w/ CVSearch in comparison to the scan-based and expert-based frameworks?
>
> Please refer to the response to **W2**.
>
> ### T216-Limitations
> > The limitations are not mentioned in the paper. One that I could see is the sometimes prohibitive for some of the applications computation burden introduced by this approach.
>
> We appreciate this valuable feedback and will explicitly discuss these limitations in the revised manuscript.

---

> > ### Author Rebuttal · Reviewer_T216 · 2026-04-03
> >
> > Thanks a lot for the provided answers. My questions were fully answered.

---

> > > ### Author Response · Authors · 2026-04-04
> > >
> > > Thank you for your positive response and for letting us know that your concerns are fully resolved. We deeply appreciate the time and effort you dedicated to reviewing our work.

---

### Decision · Program_Chairs · 2026-04-30

**Decision:**

Accept (regular)

**Comment:**

This paper presented a visual expert-assisted search approach to improving MLLM. Experimental results justified the proposed approach. All the review comments were well addressed. All the reviewers were satisifed with the rebuttals. One reviewer had a minor concern about wording, which was resolved by the further author feedback. The AC agreed with the reviewers.